# GTA1: GUI Test-time Scaling Agent

**Yan Yang**[1] **Dongxu Li** ✉[2] **Yutong Dai**[1] **Yuhao Yang**[3] **Ziyang Luo**[1]
**Zirui Zhao**[1] **Zhiyuan Hu**[1] **Junzhe Huang**[2] **Amrita Saha**[1] **Zeyuan Chen**[1]

**Ran Xu**[1] **Liyuan Pan**[4] **Caiming Xiong** ✉[1] **Junnan Li** ✉[1]

[1]Salesforce AI Research  [2]The Australian National University

[3]University of Hong Kong  [4] Yangtze Delta Region Academy  ✉ Corresponding Author

dongxuli1005@gmail.com  cxiong@salesforce.com  junnan.li@salesforce.com

## ABSTRACT

Graphical user interface (GUI) agents autonomously complete tasks across platforms (*e.g.*, Linux) by sequentially decomposing user instructions into action proposals that iteratively interact with visual elements in the evolving environment. However, two main challenges arise: i) planning (i.e., the action proposal sequence) under expansive action space, where selecting an appropriate plan is nontrivial, as many valid ones may exist; ii) accurately grounding actions in complex and high-resolution interfaces, i.e., precisely interacting with visual targets. This paper investigates the aforementioned challenges with our **G**UI **T**est-time Scaling **A**gent, namely GTA1. First, we conduct test-time scaling to select the most appropriate action proposal: at each step, multiple candidate proposals are sampled and evaluated and selected by a judge model. It trades off computation for better decision quality by concurrent sampling. Second, we propose a model that improves grounding of the selected action proposals to its corresponding visual elements. Our key insight is that reinforcement learning (RL) facilitates grounding through inherent objective alignments, rewarding successful clicks on interface elements. Experimentally, GTA1 achieves state-of-the-art performance on both grounding and agent task execution benchmarks.

## 1 INTRODUCTION

Automating task completions across diverse platforms through GUI agents represents a significant milestone toward general artificial intelligence, supporting activities from online orders to expert workflows (Yang et al., 2024). To solve a task, a GUI agent translates user instructions into multistep interactions such as action proposals consisting of clicks or keystrokes (Gou et al., 2024). This introduces a planning challenge, as multiple valid action proposal sequences may exist for the same user task. The challenge is further amplified by the high-resolution (up to 4K), complex, and hierarchical layouts of GUI (Li et al., 2025; Wu et al., 2024; Cheng et al., 2024; Xie et al., 2025), requiring accurate coordinate identification of the target interface elements. This work aims to address both challenges (i.e., planning and grounding) towards a performant GUI agent.

Formally, existing works (Yang et al., 2024; Agashe et al., 2024; 2025; Xie et al., 2025; Xu et al., 2024) often pair a GUI grounding model with a planner (*e.g.*, o3 (OpenAI, 2025b)). The planner determines an action proposal at each step, while the grounding model locates the target interface elements for interactions (*e.g.*, click areas). However, due to the inherent flexibility of user tasks, multiple feasible action proposal sequences may exist for completing the same task, some more direct and efficient than others. This makes the agent vulnerable to cascading failures, i.e., errors in early grounding or planning steps can derail the entire task. One way to avoid this is to roll out full action sequences in advance, but unlike domains such as math problem-solving, GUI environments lack a "lookahead" capability: actions often have irreversible state effects, limiting the practicality in real-world use. This raises a central question: *how can GUI agents remain robust in planning despite the lack of "lookahead" and the presence of multiple plausible action proposal sequences?*

Beyond planning, GUI grounding models predominantly rely on supervised fine-tuning (SFT) (Yang et al., 2024; Cheng et al., 2024; Gou et al., 2024; Wu et al., 2024; Xie et al., 2025), which trains

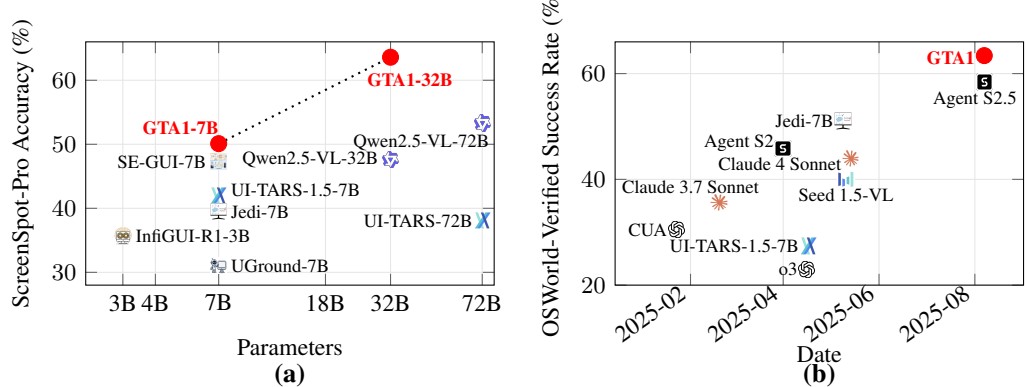

Figure 1: Comparison of **GTA1** (red dot) with state-of-the-art methods. **(a)** Grounding accuracy (%) on the ScreenSpot-Pro benchmark (Li et al., 2025) across model scales in billions (B) of parameters. **(b)** Success rate (%) on the OSWorld-Verified benchmark (Xie et al., 2024) over time.

models to predict the exact center of target interface elements. While effective in simple settings, this approach struggles to generalize to complex and high-resolution scenarios, particularly in professional interfaces (Li et al., 2025). Fundamentally, SFT introduces a misalignment with the task itself: *any coordinate within the target element constitutes a valid interaction*, yet SFT penalizes deviations from the center. It limits model flexibility, reduces robustness, and preventing the model from perceiving appropriate supervision signals.

Alternatively, inspired by DeepSeek-R1-Zero (Guo et al., 2025a), RL, particularly Group Relative Policy Optimization (GRPO) (Shao et al., 2024), has been explored in GUI grounding. Following (Guo et al., 2025a), prior works (Luo et al., 2025; Lu et al., 2025; Liu et al., 2025) formulates by performing a textual "thinking" (i. e., CoT reasoning), before predicting interaction coordinates, training with a format reward to enforce reasoning and a click reward to ensure predictions fall within the target element. Some methods further extend this approach by predicting the bounding box of the target UI element (Liu et al., 2025; Zhou et al., 2025). While improving over SFT, we ask: *is explicit "thinking" or auxiliary bounding box reward necessary for effective GUI grounding?*

We investigate the two questions by complementary strategies: i) a test-time scaling strategy for planning that works alongside a grounding model for robustly executing user tasks; ii) an RL-based grounding model that directly predicts interaction coordinates. Specifically, our test-time scaling method addresses the challenge of effective planning in an expansive action space without requiring "lookahead". Instead of committing to a single action proposal sequence, at each step, multiple candidate proposals are sampled from the planner. Then, a multimodal large language model acts as a judge to select the most contextually appropriate option. If the action is coordinate-based, the grounding model predicts the target location for accurate execution. This enables the agent to navigate complex user tasks by exploring short-term alternatives without rolling out the full sequence.

Then, we introduce a simple and straightforward GUI grounding optimization approach: the model directly predicts coordinates and receives a reward when the prediction falls within the target UI element. It effectively aligns the training objective with the task, making the method highly efficient. Despite its simplicity, this approach achieves state-of-the-art performance across diverse GUI grounding benchmarks. Interestingly, we also observe that "thinking" (i. e., reasoning) over the task object, past trajectories, and user instructions can help in dynamic environments where context evolves over time, but such strategies often fail to generalize broadly, limiting their practicality.

Overall, our contributions are summarized as follows: **i)** a comprehensive study of grounding and planning for GUI agents; **ii)** a test-time scaling strategy that improves planning robustness and reduces execution uncertainty; **iii)** a simple yet effective GUI grounding model that directly predicts interaction coordinates without requiring explicit reasoning. A performance overview is shown in Fig. 1. Our method consistently outperforms prior approaches on both GUI grounding and task completion benchmarks, providing an effective pathway toward agentic GUI behavior.

## 2 BACKGROUND

We review recent GUI agent advances, focusing on multimodal large language model-based agents, excluding non-visual approaches (*e.g.*, HTML or A11y tree-based methods). We first review the GUI grounding, and then categorize the agents into native GUI or two-stage GUI agents.

**GUI Grounding.** GUI grounding refers to the task of mapping user instructions to specific coordinates corresponding to target UI elements. Early works (Yang et al., 2024; Cheng et al., 2024; Xu et al., 2024; Gou et al., 2024) primarily focus on SFT, training models to predict the center point of the intended UI element. However, SFT does not fully align with the nature of the GUI grounding task, where any coordinate within the target element should be considered a success. As a result, SFT-based models often exhibit poor generalization (*e.g.*, on high-resolution and visually complex user interfaces (Li et al., 2025)). With the success of DeepSeek-R1-Zero (Guo et al., 2025a), RL (specifically GRPO (Shao et al., 2024)) has drawn increased attention. Many recent efforts naively replicate techniques from other domains (Luo et al., 2025; Lu et al., 2025; Liu et al., 2025), such as prompting the model to generate a "thinking" (i. e., CoT reasoning) before producing an answer, and the answer is rewarded only if the predicted coordinates fall within the target element region. This strategy overlooks an important insight: the "thinking" degrades performance in GUI grounding. A concurrent study (Zhou et al., 2025) makes similar observations, noting that CoT reasoning (i. e., "thinking") is not required for RL training in GUI grounding and may even hinder grounding accuracy. Our work further distinguishes itself in the following ways: i) we clarify that "thinking" is not necessary for GUI grounding in static environments; ii) we demonstrate that "thinking" improves grounding performance in dynamic, real-world environments when provided with past trajectories and task objectives; iii) we conduct a comprehensive study of RL-based GUI grounding across models of various scales. Going beyond prior work, we further evaluate how the model, when paired with a planner, performs in realistic and dynamic environments, a critical aspect that existing studies largely overlook.

**Two-stage GUI Agent.** One major challenge in GUI grounding is accurately locating the coordinates of UI elements intended for interaction. To address this, two-stage GUI agents modularize into planning and action, each handled by separate models (Gou et al., 2024). They usually leverage advanced reasoning models, such as GPT-4o (Hurst et al., 2024) and Claude 3.7 (Anthropic, 2025), as planners to generate an action proposal for each step from the user task instruction, using real-time UI screenshot and past trajectories as context. A separate grounding module then maps these instructions to specific UI elements, enabling the development of vision-only GUI agents (Cheng et al., 2024; Yang et al., 2024; Gou et al., 2024). While most existing work primarily focuses on GUI grounding, more complex components, such as memory management and external knowledge bases, are also being explored to enhance agent performance (Agashe et al., 2025). This paper follows the two-stage method on establishing a GUI agent.

**Native GUI Agent.** A native GUI agent completes user tasks in an end-to-end manner. Four main aspects are studied (Qin et al., 2025): i) perception, interpreting the user interface to understand the current state; ii) memory, storing knowledge and historical experiences to support making decisions; ii) planning, analyzing the task and reflecting on progress to generate action proposals; iv) action, performing atomic operations based on the action proposal to effectively progress toward the task goal. Examples of native GUI agents can be specified to CUA (Hurst et al., 2024; OpenAI, 2025b) and Claude Computer Use (Anthropic, 2025). One of the main challenges for native GUI agents lies in long-context learning. To address this, some approaches employ a sliding window mechanism (Qin et al., 2025), while others maintain a textual description of past trajectories to manage contextual information (Xu et al., 2024). In practice, end-to-end native GUI agents have demonstrated strong performance in completing user tasks, as shown by benchmarks that reflect dynamic and realistic UI environments, such as OSWorld (Xie et al., 2024). However, this paper is the first to show that a two-stage GUI agent can achieve competitive performance in such environments, challenging the assumption that end-to-end approaches are inherently superior.

## 3 METHOD

**Overview.** Our method adopts a two-stage GUI agent framework composed of a planner and a grounding model, and focuses on improving planning robustness and grounding accuracy through the following key components: i) test-time scaling for planning, which scales inference computation to effectively handle planning selection challenges in complex GUI environments; ii) grounding

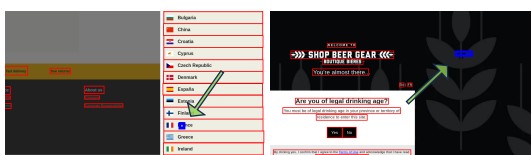

Figure 2: Overview of our GUI agent architecture. At each step, the trajectory, current UI screenshot, and user instruction are sent to a planner, which samples multiple action proposals. A multimodal large language model judge is then used to select the best candidate action proposal. When the candidate action proposal is a coordinate-based action (*e.g.*, a click), the grounding model predicts a precise interaction point on the GUI for executing the action. For non-coordinate-based actions (*e.g.*, key presses), the action can be executed directly without grounding.

model training, filtering out training samples with annotation errors to improve supervision quality, and optimizing a grounding model using RL (*e.g.*, GRPO) to directly predict coordinates without relying on any intermediate "thinking" (i. e., CoT reasoning) on the derived data.

## 3.1 TEST-TIME SCALING FOR PLANNING

At each step of executing user instructions in real-world environments, a planner is provided with the user instruction (i. e., task objective), the trajectory so far, and the current UI screenshot. Based on this context, we sample $K$ candidate action proposals, denoted as $\{p_k\}_{k=1}^K$, where $p_k$ represents a corresponding action (*e.g.*, clicking 'the blue button' or performing a keystroke).

Then, a multimodal large language model judge is used to evaluate the $K$ candidates $\{p_k\}_{k=1}^K$ based on their alignment with the user intent and the current GUI state. The judge, which can be the planner model itself, picks the best candidate action proposal $p_{k*}$ from $\{p_k\}_{k=1}^K$, allowing the agent to select the most contextually appropriate option. Once the best candidate action proposal $p_{k*}$ is selected, the grounding model $\pi(\cdot, \cdot)$ takes $p_{k*}$ and the current screenshot $s$ as the input. If $p_{k*}$ is a coordinate-based action (*e.g.*, a click), the grounding model predicts a precise interaction point on the GUI, which is then used to execute the action. For non-coordinate-based actions (*e.g.*, key presses or text input), the action can be executed directly without grounding. This process is repeated step by step until the task is completed or the agent reaches a termination condition.

By incorporating sampling, judging, and grounding into each step, our agent avoids overcommitting to suboptimal action proposals and demonstrates improved robustness in complex and dynamic GUI environments. We show an overview in Fig. 2.

## 3.2 GROUNDING

**Data Cleaning.** We leverage curated open-source datasets to train our model, *e.g.*, the Aria-UI collection (Yang et al., 2024). To train with a reward signal that verifies whether the predicted coordinates fall within the target UI element, we require a dataset that provides accurate bounding boxes for annotated interactive elements. For data points from desktop and web domains, the collection usually involves screenshots paired with accessibility tools such as A11y or HTML parsers to extract element structure and bounding box annotations. However, these bounding boxes can sometimes be

Figure 3: Examples from the Aria-UI dataset (Yang et al., 2024). The blue bounding box shows the annotation $b^{\mathrm{ann}}$, while red bounding boxes are detected by OmniParser (Lu et al., 2024). The green arrow highlights misaligned annotations, which our cleaning strategy filters out.

misaligned with the actual visual rendering due to UI animations, timing inconsistencies, or rendering delays, introducing noise into the training signal.

Therefore, to improve data quality, we apply a lightweight cleaning strategy. Given a UI screenshot $s$, we use OmniParser (Lu et al., 2024), denoted as $\mathcal{M}(\cdot)$, to detect all UI elements in $s$, resulting in a set of bounding boxes $\{b_i\} = \mathcal{M}(s)$. Each $b_i$ represents the bounding box of a detected UI element.

For a data point $s$ with an annotated bounding box $b^{\text{ann}}$, we discard the sample if the maximum Intersection over Union between $b^{\text{ann}}$ and any $b_i \in \{b_i\}$ is smaller than a predefined threshold $\tau$,

$$\max_{b_i \in \mathcal{M}(s)} \text{IoU}(b^{\text{ann}}, b_i) < \tau \,, \tag{1}$$

where $\text{IoU}(\cdot, \cdot)$ computes the overlap between two bounding boxes, defined as the area of their intersection divided by the area of their union. This helps ensure that the annotation $b^{\text{ann}}$ in the training data remains consistent with the actual visual target, thereby reducing noise caused by misaligned annotations. We show some samples in Fig. 3.

**Training.** In our RL training, we follow the GRPO framework (Guo et al., 2025a; Shao et al., 2024) to sample $N$ responses $\{o_n\}_{n=1}^N$ from the policy multimodal large language model $\pi(\cdot, \cdot)$, given a screenshot $s$ and an action proposal $p$ as input. Here, each response $o_n$ represents a pair of pixel coordinates on the screen, i.e., $o_n = (x_n, y_n)$, where $x_n$ and $y_n$ denote the horizontal and vertical positions, respectively. Unlike prior approaches, we do not prompt the model to generate a "thinking" (i.e., CoT reasoning) before producing a response. Instead, the model directly outputs the predicted coordinates, aligning more closely with the nature of the GUI grounding task.

Then, each response is evaluated by checking whether the predicted coordinate $(x_n, y_n)$ falls within the annotated bounding box $\mathbf{b}^{\text{ann}} = (x_{\min}, y_{\min}, x_{\max}, y_{\max})$. This yields a set of $N$ binary rewards $\{r_n\}_{n=1}^N$, where each reward is

$$r_n = \begin{cases} 1, & \text{if } x_{\min} \leq x_n \leq x_{\max} \text{ and } y_{\min} \leq y_n \leq y_{\max} \,, \\ 0, & \text{otherwise} \,. \end{cases} \tag{2}$$

We then normalize the rewards $\{r_n\}_{n=1}^N$ into advantages $\{A_n\}_{n=1}^N$ using Z-score normalization,

$$A_n = \frac{r_n - \frac{1}{N} \sum_{n=1}^N r_n}{\sqrt{\frac{1}{N} \sum_{n=1}^N (r_n - \frac{1}{N} \sum_{n=1}^N r_n)^2}} \,. \tag{3}$$

Finally, the model is optimized by

$$\mathcal{L} = -\frac{1}{N} \sum_{n=1}^N \min\left( \frac{\pi(o_n \mid s, p)}{\pi^{\text{old}}(o_n \mid s, p)} \cdot A_n, \text{clip}\left( \frac{\pi(o_n \mid s, p)}{\pi^{\text{old}}(o_n \mid s, p)}, 1 - \epsilon, 1 + \epsilon \right) \cdot A_n \right) \,, \tag{4}$$

where $\pi^{\text{old}}(\cdot \mid \cdot, \cdot)$ denotes the old policy, $v_n$ is the advantage associated with the prediction $o_n$, $\text{clip}(\cdot, \cdot, \cdot)$ is a clip function, and $\epsilon$ is a hyperparameter. The advantage serves as a weight, encouraging high reward predictions while suppressing low reward ones.

## 4 EXPERIMENT

**Implementation Detail.** We train our model using a mixture of dataset (Yang et al., 2024; Wu et al., 2024; Nayak et al., 2025; Kapoor et al., 2024; Li et al., 2020), applying a data cleaning threshold of $\tau = 0.3$. Our model is initialized from (Qin et al., 2025; Wang et al., 2025). When testing on the real-world dynamic environment, we use the action space from (Agashe et al., 2025) and o3 as the default planner (OpenAI, 2025b). Refer to App. B for more details.

**Dataset.** We evaluate our method on two sets of benchmarks: i) GUI Grounding, where we use ScreenSpot-Pro (Li et al., 2025), ScreenSpot-V2 (Cheng et al., 2024; Wu et al., 2024), and OSWorld-G (Xie et al., 2024) datasets, evaluating by the metric of accuracy; ii) Agent Task Execution, where we use OSWorld (Xie et al., 2024) and WindowsAgentArena (Bonatti et al., 2024) benchmarks, measuring performance by task success rate. For the OSWorld benchmark, we explore both its original release and OSWorld-Verified (i.e., an updated variant).

**Baseline.** We compare with various state-of-the-art methods: CogAgent (Hong et al., 2024), OmniParser (Lu et al., 2024), Qwen2.5-VL (Bai et al., 2025), Aria-UI (Yang et al., 2024), OS-Atlas (Wu et al., 2024), UGround (Gou et al., 2024), ShowUI (Lin et al., 2025), Aguvis (Xu et al.,

Table 1: Comparison with state-of-the-art methods on the ScreenSpot-Pro dataset (Li et al., 2025). We report the grounding accuracy (%) across various task domains, categorizing results by grounding target type: Text, Icon, and the overall average (Avg). We use '-' to denote unavailability, and '*' to mark the results evaluated by us (subject to update with improved evaluation scripts). The final average scores are highlighted in dark blue, and the best scores are in bold.

| Agent Model | Development | | | Creative | | | CAD | | | Scientific | | | Office | | | OS | | | Avg | | |
|---|---|---|---|---|---|---|---|---|---|---|---|---|---|---|---|---|---|---|---|---|---|
| | Text | Icon | Avg | Text | Icon | Avg | Text | Icon | Avg | Text | Icon | Avg | Text | Icon | Avg | Text | Icon | Avg | Text | Icon | Avg |
| *Proprietary Models* | | | | | | | | | | | | | | | | | | | | | |
| Claude 3.7 Sonnet (Anthropic, 2025) | - | - | - | - | - | - | - | - | - | - | - | - | - | - | - | - | - | - | - | - | 27.7 |
| Operator (OpenAI, 2025a) | 50.0 | 19.3 | 35.1 | 51.5 | 23.1 | 39.6 | 16.8 | 14.1 | 16.1 | 58.3 | 24.5 | 43.7 | 60.5 | 28.3 | 53.0 | 34.6 | 30.3 | 32.7 | 45.0 | 23.0 | 36.6 |
| Seed-1.5-VL (Guo et al., 2025b) | - | - | 53.8 | - | - | 59.2 | - | - | 59.0 | - | - | 61.4 | - | - | 74.8 | - | - | 60.2 | - | - | 60.9 |
| UI-TARS-1.5 (Qin et al., 2025) | - | - | 63.9 | - | - | 50.4 | - | - | 58.2 | - | - | 69.3 | - | - | 79.6 | - | - | 51.0 | - | - | 61.6 |
| *Open-Source Models* | | | | | | | | | | | | | | | | | | | | | |
| OS-Atlas-7B (Wu et al., 2024) | 33.1 | 1.4 | 17.7 | 28.8 | 2.8 | 17.9 | 12.2 | 4.7 | 10.3 | 37.5 | 7.3 | 24.4 | 33.9 | 5.7 | 27.4 | 27.1 | 4.5 | 16.8 | 28.1 | 4.0 | 18.9 |
| UI-TARS-2B (Qin et al., 2023) | 47.4 | 4.1 | 26.4 | 42.9 | 6.3 | 27.6 | 17.8 | 4.7 | 14.6 | 56.9 | 17.3 | 39.8 | 50.3 | 17.0 | 42.6 | 21.5 | 5.6 | 14.3 | 39.6 | 8.4 | 27.7 |
| Qwen2.5-VL-3B (Bai et al., 2025) | 38.3 | 3.4 | 21.4 | 40.9 | 4.9 | 25.8 | 22.3 | 6.3 | 18.4 | 44.4 | 10.0 | 29.5 | 48.0 | 17.0 | 40.9 | 33.6 | 4.5 | 20.4 | 37.8 | 6.6 | 25.9 |
| Qwen2.5-VL-7B (Bai et al., 2025) | 51.9 | 4.8 | 29.1 | 36.9 | 8.4 | 24.9 | 17.8 | 1.6 | 13.8 | 48.6 | 8.2 | 31.1 | 53.7 | 18.9 | 45.7 | 34.6 | 7.9 | 22.4 | 39.9 | 7.6 | 27.6 |
| UGround-7B (Gou et al., 2024) | - | - | 35.5 | - | - | 27.8 | - | - | 13.5 | - | - | 38.8 | - | - | 48.8 | - | - | 26.1 | - | - | 31.1 |
| UGround-72B (Gou et al., 2024) | - | - | 31.1 | - | - | 35.8 | - | - | 13.8 | - | - | 50.0 | - | - | 51.3 | - | - | 25.5 | - | - | 34.5 |
| UI-TARS-7B (Qin et al., 2023) | 58.4 | 12.4 | 36.1 | 50.0 | 9.1 | 32.8 | 20.8 | 9.4 | 18.0 | 63.9 | 31.8 | 50.0 | 63.3 | 20.8 | 53.5 | 30.8 | 16.9 | 24.5 | 47.8 | 16.2 | 35.7 |
| InfiGUI-R1-3B (Liu et al., 2025) | 51.3 | 12.4 | 32.4 | 44.9 | 7.0 | 29.0 | 33.0 | 14.1 | 28.4 | 58.3 | 20.0 | 41.7 | 65.5 | 28.3 | 57.0 | 43.9 | 12.4 | 29.6 | 49.1 | 14.1 | 35.7 |
| SE-GUI-3B (Yuan et al., 2025) | 55.8 | 7.6 | 35.1 | 47.0 | 4.9 | 29.0 | 38.1 | 12.5 | 31.8 | 61.8 | 16.4 | 43.3 | 59.9 | 24.5 | 50.9 | 40.2 | 12.4 | 25.5 | 50.4 | 11.8 | 35.9 |
| Jedi-3B (Xie et al., 2025) | 61.0 | 13.8 | 38.1 | 53.5 | 8.4 | 34.6 | 27.4 | 9.4 | 23.0 | 54.2 | 18.2 | 38.6 | 64.4 | 32.1 | 57.0 | 38.3 | 9.0 | 25.0 | 49.8 | 13.7 | 36.1 |
| GUI-G1-3B (Zhou et al., 2025) | 50.7 | 10.3 | 31.1 | 36.6 | 11.9 | 26.6 | 39.6 | 9.4 | 32.2 | 61.8 | 30.0 | 48.0 | 67.2 | 32.1 | 59.1 | 23.5 | 10.6 | 16.1 | 49.5 | 16.8 | 37.1 |
| UI-TARS-72B (Qin et al., 2023) | 63.0 | 17.3 | 40.8 | 57.1 | 15.4 | 39.6 | 18.8 | 12.5 | 17.2 | 64.6 | 20.9 | 45.7 | 63.3 | 26.4 | 54.8 | 42.1 | 15.7 | 30.1 | 50.9 | 17.5 | 38.1 |
| Jedi-7B (Xie et al., 2025) | 42.9 | 11.0 | 27.4 | 50.0 | 11.9 | 34.0 | 38.0 | 14.1 | 32.2 | 72.9 | 25.5 | 52.4 | 75.1 | 47.2 | 68.7 | 33.6 | 16.9 | 26.0 | 52.6 | 18.2 | 39.5 |
| UI-TARS-1.5-7B* (Qin et al., 2023) | 58.4 | 12.4 | 31.8 | 58.1 | 15.4 | 40.2 | 38.6 | 11.0 | 31.8 | 66.7 | 21.9 | 47.2 | 74.6 | 35.9 | 65.6 | 49.5 | 13.5 | 33.2 | 57.5 | 16.9 | 42.0 |
| Qwen2.5-VL-32B (Bai et al., 2025) | 74.0 | 21.4 | 48.5 | 61.1 | 13.3 | 41.1 | 38.1 | 15.6 | 32.6 | 78.5 | 29.1 | 57.1 | 76.3 | 37.7 | 67.4 | 55.1 | 27.0 | 42.3 | 63.2 | 22.5 | 47.6 |
| SE-GUI-7B (Yuan et al., 2025) | 68.2 | 19.3 | 44.5 | 57.6 | 9.1 | 37.2 | 51.3 | **42.2** | 42.1 | 75.0 | 28.2 | 54.7 | 78.5 | 43.4 | 70.4 | 49.5 | 25.8 | 38.8 | 63.5 | 21.0 | 47.3 |
| Qwen2.5-VL-72B (Bai et al., 2025) | - | - | 53.5 | - | - | 44.9 | - | - | 44.4 | - | - | 59.1 | - | - | 72.6 | - | - | 49.5 | - | - | 53.3 |
| OpenCUA-32B (Wang et al., 2025) | - | - | - | - | - | - | - | - | - | - | - | - | - | - | - | - | - | - | - | - | 55.3 |
| GTA1-7B | 66.9 | 20.7 | 44.5 | 62.6 | 18.2 | 44.0 | 53.3 | 17.2 | 44.4 | 76.4 | 31.8 | 57.1 | 82.5 | 50.9 | 75.2 | 48.6 | 25.9 | 38.3 | 65.5 | 25.2 | 50.1 |
| GTA1-32B | **83.1** | **37.9** | **61.2** | **72.2** | **25.9** | **52.8** | **70.1** | 31.3 | **60.5** | **84.7** | **39.1** | **65.0** | **89.3** | **64.2** | **83.5** | **76.6** | **51.7** | **65.3** | **78.9** | **38.9** | **63.6** |

2024), Jedi (Xie et al., 2025), GUI-G1 (Zhou et al., 2025), SE-GUI (Yuan et al., 2025), GUI-R1 (Luo et al., 2025), UI-R1 (Lu et al., 2025), InfiGUI-R1 (Liu et al., 2025), UI-TARS (Qin et al., 2025), Seed-1.5-VL (Guo et al., 2025b), UI-TARS-1.5 (Qin et al., 2025), o3 (OpenAI, 2025b), Claude 3.7 Sonnet (Anthropic, 2025), and Gemini-2.5 (Deepmind, 2025b).

## 4.1 GROUNDING PERFORMANCE

We compare our method with state-of-the-art approaches on the ScreenSpot-Pro (Li et al., 2025), ScreenSpot-V2 (Guo et al., 2025a; Wu et al., 2024), and OSWorld-G (Xie et al., 2024) datasets, as shown in Tab. 1, Tab. 2, and Tab. 3, respectively. Among the three benchmarks, the ScreenSpot-Pro benchmark is the most challenging, designed for high-resolution, complex, and professional GUI grounding scenarios. The ScreenSpot-V2 benchmark evaluates grounding capability across mobile, desktop, and web domains, while the OSWorld-G benchmark focuses on the Linux environment, providing a comprehensive benchmark for measuring diverse capabilities such as text matching, element recognition, layout understanding, and precise manipulation.

Our method consistently demonstrates the best performance. On the ScreenSpot-Pro (Li et al., 2025) benchmark, our 7B model outperforms much larger alternatives, for example, achieving 50.1% scores compared to 34.5% scores from UGround-72B. On the ScreenSpot-V2 benchmark, our best-performing model, GTA1-32B, achieves the same performance as the proprietary Seed-1.5-VL (Guo et al., 2025b). Similarly, on the OSWorld-G (Xie et al., 2025) benchmark, our method surpasses all state-of-the-art approaches, setting a new benchmark with a grounding accuracy of 72.2%.

Overall, our method achieves state-of-the-art performance using a simple click-based training strategy, demonstrating both robustness and effectiveness. This highlights its potential as a strong foundation for grounding models in complex GUI environments.

## 4.2 AGENT PERFORMANCE

Tab. 4 and Tab. 5 respectively compares our method with the state-of-the-art approaches on the OSWorld (Xie et al., 2024) and WindowsAgentArena (Bonatti et al., 2024) benchmarks. These

Table 2: Comparison with state-of-the-art methods on the ScreenSpot-V2 dataset (Cheng et al., 2024; Wu et al., 2024) across mobile, desktop, and web domains. We report grounding accuracy (%) categorized by target types: Text, Icon/Widget, and the overall average (Avg) highlighted in dark blue. We use '-' to denote unavailability, and '*' to mark the results evaluated by us (subject to update with improved evaluation scripts). The best scores are in bold.

| Agent Model | Mobile | | Desktop | | Web | | Avg |
| --- | --- | --- | --- | --- | --- | --- | --- |
| | Text | Icon/Widget | Text | Icon/Widget | Text | Icon/Widget | |
| *Proprietary Models* | | | | | | | |
| Operator (OpenAI, 2025a) | 47.3 | 41.5 | 90.2 | 80.3 | 92.8 | 84.3 | 70.5 |
| Claude 3.7 Sonnet (Anthropic, 2025) | - | - | - | - | - | - | 87.6 |
| UI-TARS-1.5 (Qin et al., 2025) | - | - | - | - | - | - | 94.2 |
| Seed-1.5-VL (Guo et al., 2025b) | - | - | - | - | - | - | **95.2** |
| *Open-Source Models* | | | | | | | |
| SeeClick (Guo et al., 2025b) | 78.4 | 50.7 | 70.1 | 29.3 | 55.2 | 32.5 | 55.1 |
| OmniParser-v2 (Lu et al., 2024) | 95.5 | 74.6 | 92.3 | 60.9 | 88.0 | 59.6 | 80.7 |
| Qwen2.5-VL-3B (Bai et al., 2025) | 93.4 | 73.5 | 88.1 | 58.6 | 88.0 | 71.4 | 80.9 |
| UI-TARS-2B (Qin et al., 2025) | 95.2 | 79.1 | 90.7 | 68.6 | 87.2 | 78.3 | 84.7 |
| OS-Atlas-Base-7B (Wu et al., 2024) | 95.2 | 75.8 | 90.7 | 63.6 | 90.6 | 77.3 | 85.1 |
| OS-Atlas-Base-7B (Wu et al., 2024) | 96.2 | 83.4 | 89.7 | 69.3 | 94.0 | 79.8 | 87.1 |
| Jedi-3B (Xie et al., 2025) | 96.6 | 81.5 | 96.9 | 78.6 | 88.5 | 83.7 | 88.6 |
| Qwen2.5-VL-7B (Bai et al., 2025) | 97.6 | 87.2 | 90.2 | 74.2 | 93.2 | 81.3 | 88.8 |
| UI-TARS-1.5-7B* (Qin et al., 2025) | 95.9 | 84.8 | 94.9 | 80.7 | 90.6 | 86.2 | 89.7 |
| UI-TARS-72B (Qin et al., 2025) | 94.8 | 86.3 | 91.2 | 87.9 | 91.5 | 87.7 | 90.3 |
| UI-TARS-7B (Qin et al., 2025) | 96.9 | 89.1 | 95.4 | 85.0 | 93.6 | 85.2 | 91.6 |
| Jedi-7B (Xie et al., 2025) | 96.9 | 87.2 | 95.9 | 87.9 | 94.4 | 84.2 | 91.7 |
| Qwen2.5-VL-32B* (Bai et al., 2025) | 98.3 | 86.7 | 94.3 | 83.6 | 93.6 | 89.7 | 91.9 |
| Qwen2.5-VL-72B* (Bai et al., 2025) | 99.0 | 90.1 | 96.4 | 87.1 | **96.6** | **90.6** | 94.0 |
| OpenCUA-32B (Wang et al., 2025) | - | - | - | - | - | - | 93.4 |
| GTA1-7B | 99.0 | 88.6 | 94.9 | 89.3 | 92.3 | 86.7 | 92.4 |
| GTA1-32B | **99.7** | **90.5** | **99.0** | **94.3** | 95.7 | 90.1 | **95.2** |

Table 3: Performance comparison of state-of-the-art models on the OSWorld-G (Xie et al., 2025) dataset. We report grounding accuracy (%) categorized by different capabilities, along with the overall average (Avg) highlighted in dark blue. We use '-' to denote unavailability, and '*' to mark the results evaluated by us (subject to update with improved evaluation scripts). For methods not tagged with [†], the results are based on refined grounding instructions. The best scores are in bold.

| Agent Model | Text Matching | Element Recognition | Layout Understanding | Fine-grained Manipulation | Refusal | Avg |
| --- | --- | --- | --- | --- | --- | --- |
| *Proprietary Models* | | | | | | |
| Operator[†] (OpenAI, 2025a) | 51.3 | 42.4 | 46.6 | 31.5 | 0.0 | 40.6 |
| Operator (OpenAI, 2025a) | - | - | - | - | - | 57.8 |
| Gemini-2.5-Pro (Deepmind, 2025a) | 59.8 | 45.5 | 49.0 | 33.6 | **38.9** | 45.2 |
| Gemini-2.5-Pro (Deepmind, 2025a) | - | - | - | - | - | 57.5 |
| Seed1.5-VL[†] (Guo et al., 2025b) | 73.9 | 66.7 | 69.6 | 47.0 | 18.5 | 62.9 |
| *Open-Source Models* | | | | | | |
| Qwen2.5-VL-3B[†] (Bai et al., 2025) | 41.4 | 28.8 | 34.8 | 13.4 | 0.0 | 27.3 |
| OS-Atlas-7B[†] (Wu et al., 2024) | 44.1 | 29.4 | 35.2 | 16.8 | 7.4 | 27.7 |
| Qwen2.5-VL-7B[†] (Bai et al., 2025) | 45.6 | 32.7 | 41.9 | 18.1 | 0.0 | 31.4 |
| UGround-7B[†] (Gou et al., 2024) | 51.3 | 40.3 | 43.5 | 24.8 | 0.0 | 36.4 |
| Aguvis-7B[†] (Xu et al., 2024) | 55.9 | 41.2 | 43.9 | 28.2 | 0.0 | 38.7 |
| UI-TARS-7B[†] (Qin et al., 2025) | 60.2 | 51.8 | 54.9 | 35.6 | 0.0 | 47.5 |
| Qwen2.5-VL-32B (Bai et al., 2025) | 57.9 | 70.2 | 73.8 | 49.2 | 0.0 | 59.6 |
| Jedi-3B[†] (Xie et al., 2025) | 67.4 | 53.0 | 53.8 | 44.3 | 7.4 | 50.9 |
| Jedi-3B (Xie et al., 2025) | - | - | - | - | - | 61.0 |
| Jedi-7B[†] (Xie et al., 2025) | 65.9 | 55.5 | 57.7 | 46.9 | 7.4 | 54.1 |
| Jedi-7B (Xie et al., 2025) | - | - | - | - | - | 63.8 |
| UI-TARS-72B[†] (Qin et al., 2025) | **69.4** | 60.6 | 62.9 | 45.6 | 0.0 | 57.1 |
| Qwen2.5-VL-72B (Bai et al., 2025) | 52.6 | 74.6 | 74.7 | 55.3 | 0.0 | 62.2 |
| UI-TARS-1.5-7B*[†] (Qin et al., 2025) | 36.8 | 62.7 | 62.2 | 50.8 | 0.0 | 52.8 |
| UI-TARS-1.5-7B* (Qin et al., 2025) | 52.6 | 75.4 | 72.4 | 66.7 | 0.0 | 64.2 |
| OpenCUA-32B[†] (Wang et al., 2025) | - | - | - | - | - | 59.6 |
| OpenCUA-32B* (Wang et al., 2025) | 63.2 | 79.9 | **84.9** | 62.1 | 7.4 | 70.2 |
| GTA1-7B[†] | 42.1 | 65.7 | 62.7 | 56.1 | 0.0 | 55.1 |
| GTA1-7B | 63.2 | 82.1 | 74.2 | **70.5** | 0.0 | 67.7 |
| GTA1-32B[†] | 63.2 | 78.4 | 73.3 | 65.2 | 0.0 | 65.2 |
| GTA1-32B | 63.2 | **83.6** | 84.4 | **70.5** | 0.0 | **72.2** |

benchmarks evaluate agents on completing user tasks in real-world desktop applications, such as web browsers and office softwares. We assess various scales of our grounding model using o3 and GPT-5 as both the planner and the test-time scaling judge, forming the GTA1 agent series. Our method consistently finds the best performance across all benchmarks. For example, GTA1-

Table 4: Comparison with state-of-the-art methods on the OSWorld and OSWorld-Verified (Xie et al., 2024) benchmarks. A dash ('-') indicates unavailable results, the second column shows the number of steps, and success rate success rate (%) is reported. The best scores are in bold.

| Agent Model | Step | OSWorld | OSWorld-Verified |
|---|---|---|---|
| *Proprietary Models* | | | |
| Claude 3.7 Sonnet (Anthropic, 2025) | 100 | 28.0 | - |
| OpenAI CUA 4o (OpenAI, 2025b) | 200 | 38.1 | - |
| UI-TARS-1.5 (Qin et al., 2025) | 100 | 42.5 | 41.8 |
| OpenAI CUA o3 (OpenAI, 2025b) | 200 | 42.9 | - |
| *Open-Source Models* | | | |
| Aria-UI w/ GPT-4o (Yang et al., 2024) | 15 | 15.2 | - |
| Aguvis-72B w/ GPT-4o (Xu et al., 2024) | 15 | 17.0 | - |
| UI-TARS-72B-SFT (Qin et al., 2025) | 50 | 18.8 | - |
| Agent S w/ Claude-3.5-Sonnet (Agashe et al., 2024) | 15 | 20.5 | |
| Agent S w/ GPT-4o (Agashe et al., 2024) | 15 | 20.6 | - |
| o3 (OpenAI, 2025b) | 100 | 20.1 | 23.0 |
| UI-TARS-72B-DPO (Qin et al., 2025) | 15 | 22.7 | - |
| UI-TARS-72B-DPO (Qin et al., 2025) | 50 | 24.6 | - |
| UI-TARS-1.5-7B (Qin et al., 2025) | 100 | 26.9 | 27.4 |
| Jedi-7B w/ o3 (Xie et al., 2025) | 100 | - | 51.0 |
| Jedi-7B w/ GPT-4o (Xie et al., 2025) | 100 | 27.0 | - |
| Agent S2 w/ Claude-3.7-Sonnet (Agashe et al., 2025) | 50 | 34.5 | - |
| Agent S2 w/ Gemini-2.5-Pro (Agashe et al., 2025) | 50 | 41.4 | 45.8 |
| Agent S2.5 w/ o3(Agashe et al., 2025) | 100 | - | 56.0 |
| Agent S2.5 w/ GPT-5(Agashe et al., 2025) | 100 | - | 58.4 |
| CoAct-1 w/o3 & o4mini & OpenAI CUA 4o (Song et al., 2025) | 150 | - | 60.8 |
| GTA1-7B w/ o3 | 100 | **45.2** | 53.1 |
| GTA1-7B w/ GPT-5 | 100 | - | 61.0 |
| GTA1-32B w/ o3 | 100 | - | 55.4 |
| GTA1-32B w/ GPT-5 | 100 | - | **63.4** |

Table 5: Comparisons on WindowsAgentArena (Bonatti et al., 2024) benchmarks. We report success rate (%) for evaluations. The best scores are in bold.

| Agent Model | Step | Success Rate |
|---|---|---|
| Kimi-VL (Team et al., 2025) | 15 | 10.4 |
| WAA (Bonatti et al., 2024) | - | 19.5 |
| Jedi-7B (Xie et al., 2025) | 100 | 33.7 |
| GTA1-7B w/ o3 | 100 | 47.9 |
| GTA1-7B w/ GPT-5 | 100 | 49.2 |
| GTA1-32B w/ o3 | 100 | **51.2** |
| GTA1-32B w/ GPT-5 | 100 | 50.6 |

Table 6: Ablation of optimization rewards. We study three types of rewards used to guide model learning: click reward (i. e., whether the prediction falls within the target element bounding box), IoU reward (i. e., enforcing predictions of the bounding box of the target element), and format reward (i. e., enforcing "thinking" before predictions).

| Click Reward | IoU Reward | Format Reward | ScreenSpot-Pro | ScreenSpot-V2 | OSWorld-G |
|---|---|---|---|---|---|
| ✓ | ✓ | ✓ | 44.5 | 89.3 | 59.9 |
| ✓ | ✓ | | 42.2 | 89.2 | 59.2 |
| ✓ | | ✓ | 46.9 | 93.2 | 67.0 |
| ✓ | | | 50.1 | 92.4 | 67.7 |

7B achieves the highest task success rate of 45.2% on the OSWorld benchmark, outperforming all state-of-the-art methods. It is worth highlighting that, even when using the o3 planner, GTA1-7B significantly outperforms its native agent variant, CUA o3, while operating with a shorter execution horizon (i. e., 45.2% from our method with a 100-step horizon *vs.* 42.9% from CUA o3 with a 200-step horizon (OpenAI, 2025a)). The strong performance of the GTA1 agent demonstrates its effectiveness in handling complex and real-world user tasks across diverse scenarios.

## 4.3 DISCUSSION AND ABLATION

**Click Reward Works the Best.** We explore different optimization objectives for GTA1, focusing on widely studied rewards such as the format reward (i. e., enforcing "thinking") and IoU rewards (i. e., encouraging accurate bounding box predictions for the target element) in Tab. 6. We evaluate three settings: optimizing the model using the format reward, IoU reward, and click reward (Eq. (2)); using the IoU reward and click reward; applying the format reward and click reward. The three settings achieve performance of 44.5%/42.2%/46.9%, 89.3%/89.2%/93.2%, and 59.9%/59.2%/67.0% respectively on the ScreenSpot-Pro (Li et al., 2025), ScreenSpot-V2 (Cheng et al., 2024; Wu et al., 2024), and OSWorld-G (Xie et al., 2024) benchmarks. However, all of the settings generally underperform compared to using the click reward alone (except on the ScreenSpot-V2 benchmark), which yields the accuracies of 50.1%, 92.4%, and 67.7% on the three benchmarks.

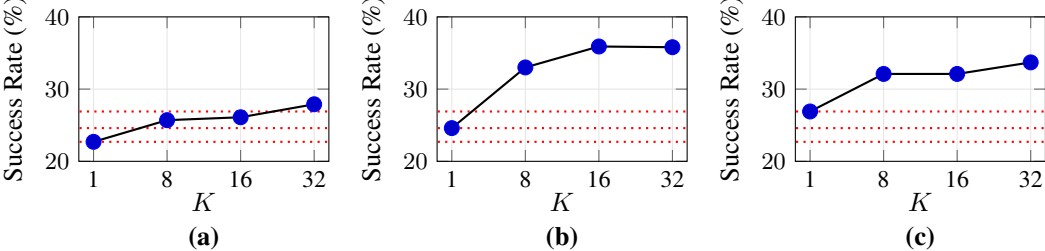

Figure 4: Scalability of the number $K$ of action proposals in test-time scaling. The task execution steps in **(a)**, **(b)**, and **(c)** are 15, 50, and 100, respectively. We vary $K$ over {1, 8, 16, 32}, and measure performance using task success rate (%). When $K = 1$, it degrades to the setting where the test-time scaling strategy is not used. We color the performance of $K = 1$ over {15, 50, 100} steps on the OSWorld benchmark with red dotted lines.

**"thinking" Benefits Grounding in Dynamic Environment Only.** Across various benchmarks, we observe minimal performance differences between grounding models trained with and without "thinking". However, they often succeed on different samples, likely due to training instability rather than systematic reasoning gains. We find that "thinking" can be effective in dynamic environments such as the AndroidWorld benchmark (Rawles et al., 2024), where the model is provided with the task object, past trajectories, and the user instruction. For example, by training an in-domain 7B model based on (Bai et al., 2025) using the AndroidControl dataset (Li et al., 2024), using or not using "thinking" have similar grounding performance on the AndroidControl test fold. However, the task success rate on the AndroidWorld benchmark increased from 39% to 44% when using "thinking". This improvement is attributed to the increased complexity of the textual prompts (i. e., combination of task object, past trajectories, and the user instruction), which encourages the model to engage in "thinking" when operating under challenging and dynamic conditions.

**Test-time Scaling Generalizes Well.** We demonstrate the generalization capability of our test-time scaling strategies through two sets of experiments. First, with $K = 8$ action proposals, increasing the horizon from 50 to 100 steps on the OSWorld benchmark (Xie et al., 2024) improves the success rate from 43.4% to 45.2%, indicating robustness across varying execution lengths. In comparison, the baseline performance with $K = 1$ is 41.3% and 43.4% for the 50-step and 100-step horizons, respectively. Second, we show that our test-time scaling strategies generalize to other agents, as evidenced by the scalability of $K$ on UI-TARS-1.5-7B (Qin et al., 2025), shown in Fig. 4, while we show its token usage in Fig. 5. We assess performance with 15-, 50-, and 100-step horizons in Fig. 4 (a), Fig. 4 (b), and Fig. 4 (c) respectively, by varying $K$ over {1, 8, 16, 32}. The red dotted lines mark the baseline performance of UI-TARS-1.5-7B without any test-time scaling (i. e., $K = 1$) at each horizon on the OSWorld benchmark. Our test-time scaling strategy consistently boosts performance, yielding two main insights: i) with our test-time scaling, UI-TARS-1.5-7B executed for only 15 steps and $K = 32$ already outperforms the baseline that executes for 100 steps without scaling. Since the $K$ candidate action proposals are sampled concurrently, this also cuts wall-clock time substantially; ii) the greatest overall gain occurs with a 50-step horizon. Using 15-step horizon is occasionally insufficient to complete certain tasks, whereas 100-step horizon provide excessive slack, diluting the benefit of additional steps. Moreover, we present the qualitative comparisons of UI-TARS-1.5-7B with and without test-time scaling strategies in App. A, highlighting the effectiveness of the test-time strategy in making the agent highly susceptible to cascading failures.

## 5 CONCLUSION AND LIMITATION

This paper investigates two key challenges toward building intelligent GUI agents: selecting effective plans and precise grounding in complex interfaces. We address these challenges with two strategies. First, to improve task planning, we introduce a scalable test-time strategy that concurrent samples multiple action proposals at each step and use a multimodal large language model judge to select the most suitable one. Second, we propose a grounding model, which employs a simple RL-based optimization approach that directly rewards successful clicks on target elements, bypassing the explicit "thinking" required by prior methods. Overall, GTA1 achieves state-of-the-art performance on standard grounding benchmarks and demonstrates robust behavior when integrated with a

planner and our test-time scaling strategy for user task execution in GUI environment. However, our approach has limitations. Although it achieves the highest accuracy on the challenging ScreenSpot-Pro benchmark, it still struggles in certain scenarios. For example, similar to prior work, our grounding model has difficulty in selecting custom foregrounds and backgrounds when applied to image editing with GMIP. We hope this work inspires further research on GUI agents.

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

# Table of Contents in Appendix

# A ADDITIONAL RESULT

## A.1 QUALITATIVE COMPARISONS

We present qualitative comparisons of UI-TARS-1.5-7B (Qin et al., 2025) with and without test-time scaling strategies in Fig. 6 and Fig. 8. As shown, without the test-time strategy, errors in early grounding or planning stages can propagate and derail the entire task execution, making the agent highly susceptible to cascading failures.

## A.2 TOKEN USAGE

We show the performance with respect to the average number of tokens used per task for test-time scaling with UI-TARS-1.5-7B (Qin et al., 2025) in Fig. 5.

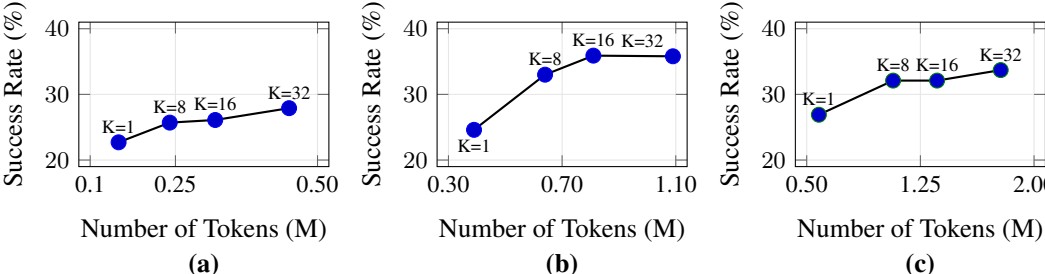

Figure 5: Average number of tokens used per task (in millions) versus task success rate across different numbers $K$ of action proposals in test-time scaling. Subfigures **(a)**, **(b)**, and **(c)** correspond to task execution lengths of 1, 15, 50, and 100 steps, respectively. We vary $K \in \{1, 8, 16, 32\}$ and report performance using task success rate (%) together with the average number of tokens consumed (in millions). When $K = 1$, it degrades to the setting where the test-time scaling strategy is not used.

## A.3 FAILURE CASE

We present several failure cases of test-time scaling in Fig. 7. In these cases, all generated action proposals $\{p\}$ fail to produce the correct next step toward completing the task, ultimately causing the judge to select an incorrect action. Specifically, in the top row, the model fails to recognize that setting the number of the search result display in Google is infeasible. In the bottom row, the method lacks the knowledge to correctly insert a note in LibreOffice Impress and instead proposes an incorrect random action.

# B MODEL TRAINING DETAILS

Our models are initialized from UI-TARS-1.5-7B (Qin et al., 2025) and OpenCUA-32B (Wang et al., 2025). We use learning rates of $10^{-6}$ and $10^{-5}$, respectively, rolling out $N = 8$ responses per input during training. The batch size is set to 256, and the 7B and 32B models typically converge after approximately 250 iterations. The training parameters are summarized below (Tab. 7).

During training, all images are resized to make their dimensions are divisible by 28 (Bai et al., 2025). The predictions are then retrieved, and the ground-truth bounding boxes are scaled by the same ratio for reward calculation. The 7B and 32B models are trained on 16 H100 and 32 H200 GPUs, taking approximately 2 and 1 days, respectively.

When evaluating on real-world dynamic environment benchmarks (i. e., OS-World (Xie et al., 2024) and WindowAgentArena (Bonatti et al., 2024)), we use the action space from (Agashe et al., 2025) and employ o3 and GPT-5 as planners (OpenAI, 2025b). By default, $K = 8$ action proposals are sampled at each step for selection.

The training data are sourced from Aria-UI-Web (Yang et al., 2024), OmniACT (Kapoor et al., 2024), UI Vision (Nayak et al., 2025), Widget Caption (Li et al., 2020), and OS-Altas-Desktop

**User Instruction**: Search for a one way flight from Dublin to Vienna on 10th next month for 2 adults.

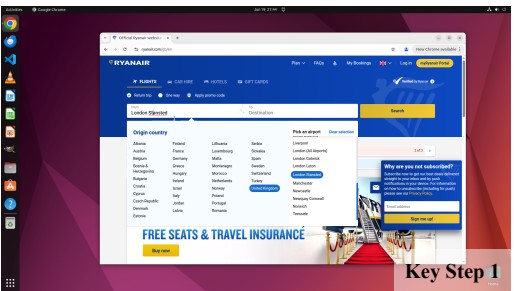
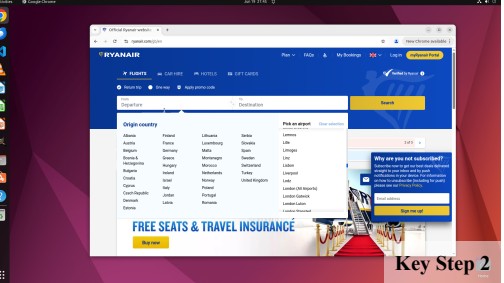

**Key Step 1: Click the "London Stansted"**

**Key Step 2: Clear the text field**
(After multiple attempts of modifying the text to "Dublin")

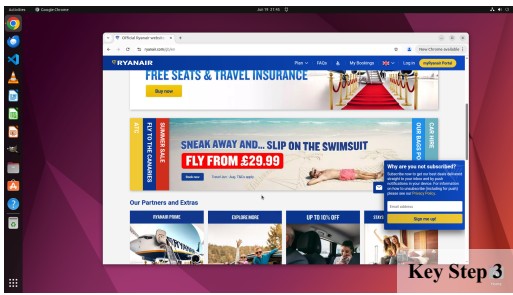
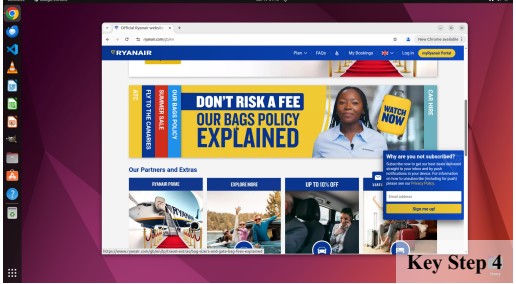

**Key Step 3: Scroll down to find the flight**

**Key Step 4: Scroll up to find the flight**
(Fail to locate ticket by scrolling)

(a) w/o Test-time Scaling

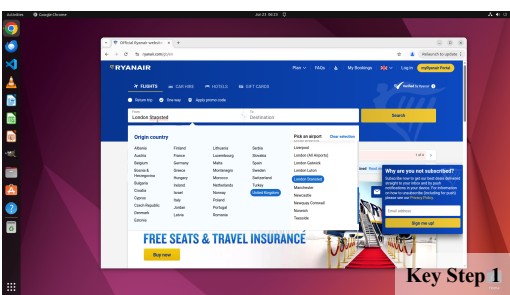
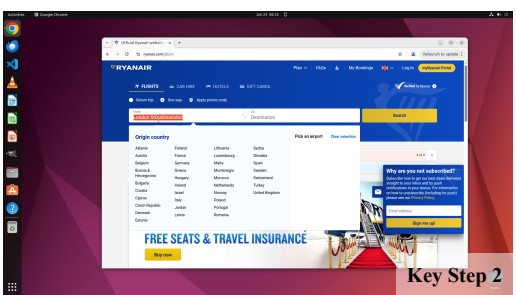

**Key Step 1: Click the "London Stansted"**

**Key Step 2: Attempt to modify the "From" field to "Dublin"**

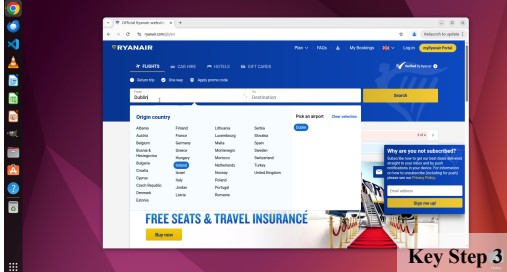
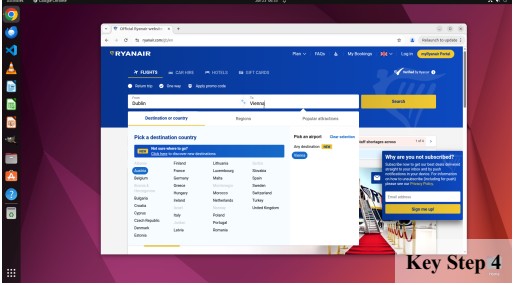

**Key Step 3: Type "Dublin" to the "From" field**

**Key Step 4: Keep modifying the search information**

(b) w/ Test-time Scaling

Figure 6: Example trajectories improved by our test-time scaling strategy. We show key steps of completing the user task using UI-TARS-1.5-7B. **(a)** Without our test-time scaling strategy, UI-TARS-1.5-7B shifts its action proposal from modifying the search field to scrolling the page to find the ticket. This occurs due to early planning and grounding errors in the "From" field. **(b)** With our strategy, it consistently modifies the search information to complete the task.

**User Instruction**: Could you please change the number of search results displayed on one page to 50? I find that having more results visible at once significantly enhances my research efficiency, as it reduces the need to constantly click through multiple pages.

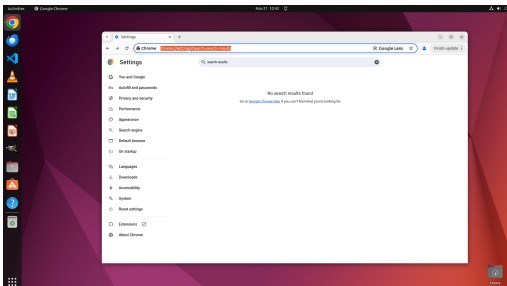

$p_1$: Previous attempts failed to find the desired settings, and I'm already in the address bar; it's time for a change of approach. Since it's a matter for Google's own search box, why not just go to their website? Typing google.com into the address bar should be the most direct way.

$p_2$: This search function seems a bit weak; I can't find an option to adjust the number of results displayed. I remember Chrome's settings page has a "Performance" option, which should involve some interface display settings. I'll check the left-hand menu first; maybe I can find the function I need there.

$p_3$: ...

**User Instruction**: On page 2, add a note same to the title into the slide. Make the font of title bold.

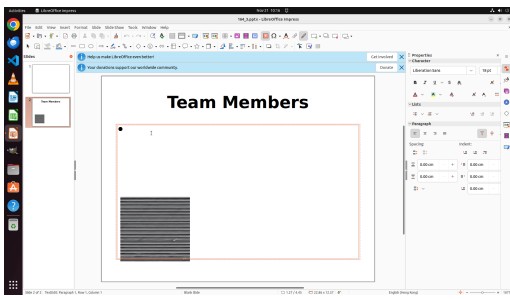

$p_1$: I noticed a "Click to add Text" area in the center of the slide, waiting for input. The key to completing the task is to add the title "Team Members" itself as a note. The cursor is now in the right position; I need to directly type this text to maintain consistency with the title.

$p_2$: I just clicked the input box to write a note, but that doesn't seem right. Looking at the menu bar, it seems more appropriate to add notes using the Insert function. Let me click the "Insert" option at the top to start this operation again.

$p_3$: ...

Figure 7: Failure cases. In these examples, all action proposals $\{p\}$ fail to contain the correct step needed to progress toward the task instruction, leading the judge to select an incorrect action since no proposal is valid. In the top row, the model fails to recognize that the task is infeasible. In the bottom row, the model lacks the required knowledge on how to add a note in LibreOffice Impress.

Table 7: Training parameters for GTA1 models.

| Parameter | GTA1-7B | GTA1-32B |
|---|---|---|
| Base Model | UI-TARS-1.5-7B (Qin et al., 2025) | OpenCUA-32B (Wang et al., 2025) |
| Learning Rate | $10^{-6}$ | $10^{-5}$ |
| Optimizer | AdamW | AdamW |
| Max Model Length | 32768 | 32768 |
| Model Type | bfloat16 | bfloat16 |
| Max Gradient Clip Norm | 1 | 1 |
| Optimization Iterations | 250 | 250 |
| $N$ (Rollout per Input) | 8 | 8 |
| Value for clipping $\epsilon$ | $2 \times 10^{-1}$ | $2 \times 10^{-1}$ |
| Rollout Temperature | 1 | 1 |
| Rollout Max Response Length | 32 | 128 |
| Image Processor Max Pixels | 12356789 | 12356789 |
| Training GPUs | 16 H100 | 32 H200 |

(Wu et al., 2024). 70K datasets are sampled from training, and the retained dataset is sampled as summarized in the table below to form our 70K training dataset.

Table 8: Distribution of datasets and domains in our training set.

| Domain | Dataset | Percentage |
|---|---|---|
| | OS-Atlas-Desktop | 57.3 |
| OS | OmniACT | 2.6 |
| | UI Vision | 5.7 |
| Web | Aria-UI-Web | 14.2 |
| Mobile | Widget Caption | 20.2 |

**User Instruction**: Hey, I need a quick way back to this site. Could you whip up a shortcut on my desktop for me?

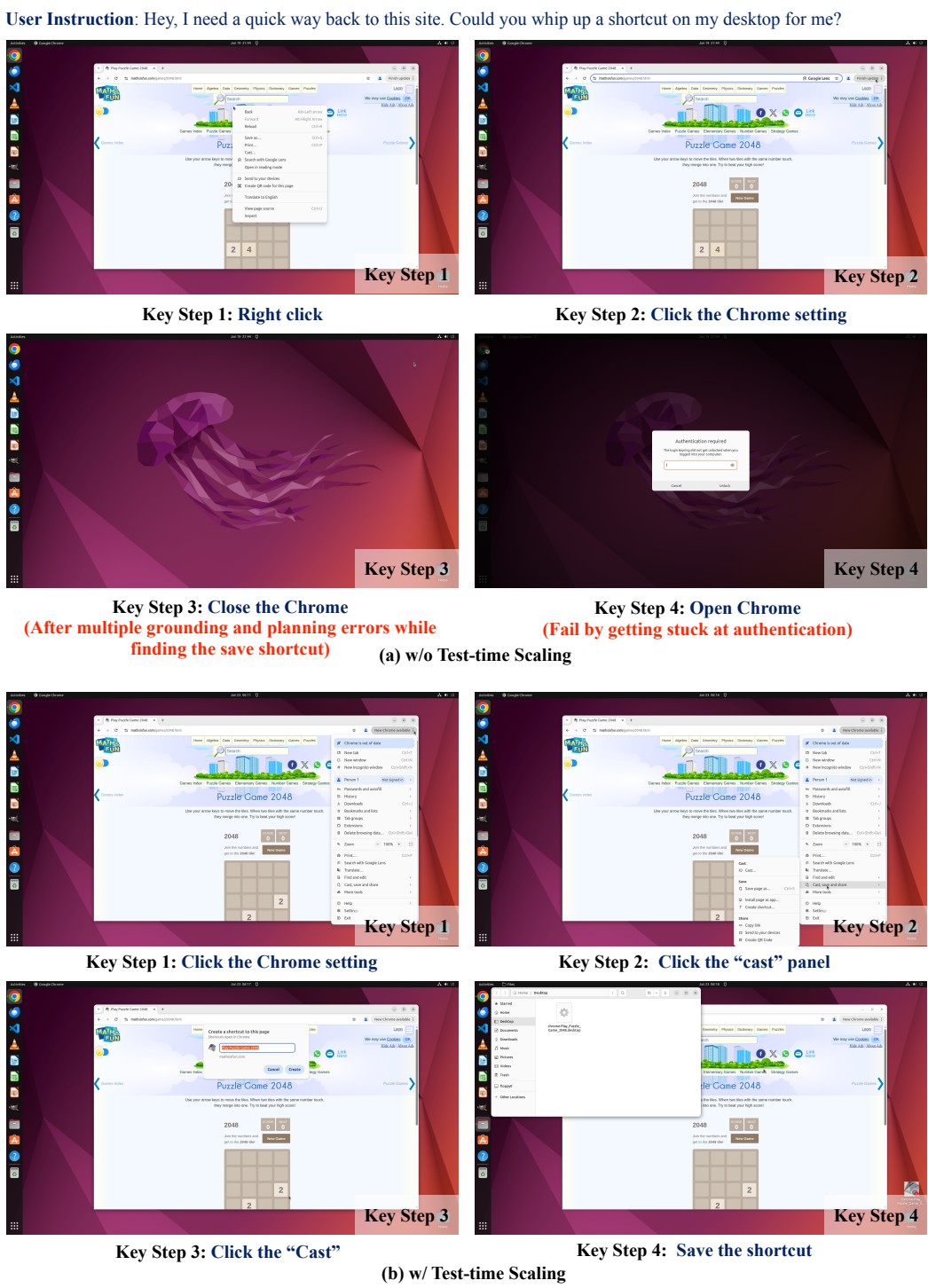

Figure 8: Example trajectories improved by our test-time scaling strategy. We show key steps of completing the user task using UI-TARS-1.5-7B. **(a)** Without our strategy, UI-TARS-1.5-7B attempts to save a shortcut by closing Chrome, encountering authentication, and getting stuck. This results from early planning and grounding errors in locating the shortcut panel. **(b)** With our strategy, it focuses on opening Chrome settings and successfully completes the task.

## C    EVALUATION DETAILS

We evaluate our method for GUI Grounding on the ScreenSpot-V2 (Cheng et al., 2024; Wu et al., 2024), ScreenSpot-Pro (Li et al., 2025), and OSWorld-G (Xie et al., 2024) benchmarks, and for Agent Task Execution on the OSWorld (Xie et al., 2024) and WindowsAgentArena (Bonatti et al., 2024) benchmarks. Detailed descriptions are provided below.

### C.1    SCREENSPOT-V2

The ScreenSpot-V2 benchmark (Cheng et al., 2024; Wu et al., 2024) is an extension of ScreenSpot (Cheng et al., 2024), correcting and re-annotating 11.32% of incorrect samples (*e.g.*, fixing spelling errors and incorrect bounding boxes). It assesses grounding ability across three domains: mobile, desktop, and web. The native image resolution and paired instructions are fed into our model for grounding, using a single GPU with bfloat16 precision.

### C.2    SCREENSPOT-PRO

The ScreenSpotPro benchmark (Li et al., 2025) focuses on professional high-resolution (up to $3840 \times 2160$) computer use and are mainly categorized into Development, Creative, CAD, Scientific, Office, and OS domains, spanning 23 applications. The paired instructions are typically concise and instruction-level, further increasing the challenge. Both our 7B and 32B models can perform inference efficiently on a single 80GB GPU with bfloat16 precision.

### C.3    OSWORLD-G

The OSWorld-G benchmark (Xie et al., 2025) is curated with finely annotated samples across five task types: text matching, element recognition, layout understanding, precise manipulation, and refusal. In addition to instruction-level annotations, it provides fine-grained annotations for each example, rephrasing the original instructions to explicitly decompose the GUI knowledge required to complete the task. We report performance on both annotation types. Similarly, our 7B and 32B models perform inference efficiently on a single 80GB GPU with bfloat16 precision.

### C.4    OSWORLD

The OSWorld benchmark (Xie et al., 2024), based on the Ubuntu operating system, originally contains computer tasks spanning both web and desktop applications in open domains. It has been further refined by improving environment stability and evaluation functions, resulting in OSWorld-Verified (Xie et al., 2024). We evaluate both versions using o3 and GPT-5 (OpenAI, 2025b) as planners, while serving our 7B and 32B models with the `vllm` codebase (Kwon et al., 2023) for grounding. All evaluations are performed using 48 Docker instances (i. e., virtual environments) and 8 served models (Kwon et al., 2023).

### C.5    WINDOWSAGENTARENA

The WindowAgentArena benchmark (Bonatti et al., 2024) is built on the Windows operating system and focuses on commonly used applications, tools, and web browsers. Similar to the OSWorld evaluations, we evaluate it using o3 and GPT-5 (OpenAI, 2025b) as planners, running under 8 Docker instances (i. e., virtual environments) with 8 served models (Kwon et al., 2023).

## D    PROMPT

We present the system prompts used to evaluate our models, categorized into grounding and planning prompts.

### D.1    GROUNDING PROMPT

We present the GTA1-7B and GTA1-32B prompt, respectively in Tab. 9 and Tab. 10.

Table 9: The system prompt used for GUI grounding with GTA1-7B.

```
You are an expert UI element locator. Given a GUI image and
a user's element description, provide the coordinates of the
specified element as a single (x,y) point. The image
resolution is height {height} and width {width}. For
elements with area, return the center point.

Output the coordinate pair exactly:
(x,y)
```

Table 10: The system prompt used for GUI grounding with GTA1-32B.

```
You are a GUI agent. You are given a task and a screenshot
of the screen. You need to perform a series of pyautogui
actions to complete the task.
```

## D.2 PLANNING PROMPT

We present the system prompt tuned for o3 and GPT-5 as the planner in Tab. 11 and Tab. 12.

Table 11: The system prompt employed for planning task executions with o3.

```
You are an agent which follow my instruction and perform
desktop computer tasks as instructed.
You have good knowledge of computer and good internet
connection and assume your code will run on a computer for
controlling the mouse and keyboard.
You are on Ubuntu operating system and the resolution of the
screen is 1920x1080.
For each step, you will get:
- An observation of an image, which is the screenshot of the
computer screen and you will predict the action of the
computer based on the image.
- Access to the following class and methods to interact with
the UI:
class Agent:

    def click(self, instruction: str, num_clicks: int = 1,
    button_type: str = 'left', hold_keys: List = []):
    '''Click on the element
        Args:
            instruction:str, decribe the element you want to
            interact with in detail including the visual
            description and function description. And make
            it clear and concise. For example you can
            describe what the element looks like, and what
            will be the expected result when you interact
            with it.
```

```
            num_clicks:int, number of times to click the
            element
            button_type:str, which mouse button to press can
            be "left", "middle", or "right"
            hold_keys:List, list of keys to hold while
            clicking
        '''

    def done(self, return_value: Union[Dict, str, List,
    Tuple, int, float, bool, NoneType] = None):
    '''End the current task with a success and the required
    return value'''

    def drag_and_drop(self, starting_description: str,
    ending_description: str, hold_keys: List = []):
    '''Drag from the starting description to the ending
    description
        Args:
            starting_description:str, a very detailed
            description of where to start the drag action.
            This description should be at least a full
            sentence. And make it clear and concise.
            ending_description:str, a very detailed
            description of where to end the drag action.
            This description should be at least a full
            sentence. And make it clear and concise.
            hold_keys:List list of keys to hold while
            dragging
        '''

    def fail(self):
    '''End the current task with a failure, and replan the
    whole task.'''

    def highlight_text_span(self, starting_phrase: str,
    ending_phrase: str):
    '''Highlight a text span between a provided starting
    phrase and ending phrase. Use this to highlight words,
    lines, and paragraphs.
        Args:
            starting_phrase:str, the phrase that denotes the
            start of the text span you want to highlight. If
            you only want to highlight one word, just pass
            in that single word.
            ending_phrase:str, the phrase that denotes the
            end of the text span you want to highlight. If
            you only want to highlight one word, just pass
            in that single word.
        '''

    def hold_and_press(self, hold_keys: List, press_keys:
    List):
    '''Hold a list of keys and press a list of keys
        Args:
            hold_keys:List, list of keys to hold
            press_keys:List, list of keys to press in a
            sequence
```

```
        '''

    def hotkey(self, keys: List):
    '''Press a hotkey combination
        Args:
            keys:List the keys to press in combination in a
            list format (e.g. ['ctrl', 'c'])
        '''

    def open(self, app_or_filename: str):
    '''Open any application or file with name
    app_or_filename. Use this action to open applications or
    files on the desktop, do not open manually.
        Args:
            app_or_filename:str, the name of the application
            or filename to open
        '''

    def scroll(self, instruction: str, clicks: int, shift:
    bool = False):
    '''Scroll the element in the specified direction
        Args:
            instruction:str, a very detailed description of
            which element to enter scroll in. This
            description should be at least a full sentence.
            And make it clear and concise.
            clicks:int, the number of clicks to scroll can
            be positive (up) or negative (down).
            shift:bool, whether to use shift+scroll for
            horizontal scrolling
        '''

    def set_cell_values(self, cell_values: Dict[str, Any],
    app_name: str, sheet_name: str):
    '''Use this to set individual cell values in a
    spreadsheet. For example, setting A2 to "hello" would be
    done by passing {"A2": "hello"} as cell_values. The
    sheet must be opened before this command can be used.
        Args:
            cell_values: Dict[str, Any], A dictionary of
            cell values to set in the spreadsheet. The keys
            are the cell coordinates in the format "A1",
            "B2", etc.
                Supported value types include: float, int,
                string, bool, formulas.
            app_name: str, The name of the spreadsheet
            application. For example, "Some_sheet.xlsx".
            sheet_name: str, The name of the sheet in the
            spreadsheet. For example, "Sheet1".
        '''

    def switch_applications(self, app_code):
    '''Switch to a different application that is already open
        Args:
            app_code:str the code name of the application to
            switch to from the provided list of open
            applications
```

```
        '''

    def type(self, element_description: Optional[str] =
    None, text: str = '', overwrite: bool = False, enter:
    bool = False):
    '''Type text into a specific element
        Args:
            element_description:str, a detailed description
            of which element to enter text in. This
            description should be at least a full sentence.
            text:str, the text to type
            overwrite:bool, Assign it to True if the text
            should overwrite the existing text, otherwise
            assign it to False. Using this argument clears
            all text in an element.
            enter:bool, Assign it to True if the enter key
            should be pressed after typing the text,
            otherwise assign it to False.
        '''

    def wait(self, time: float):
    '''Wait for a specified amount of time
        Args:
            time:float the amount of time to wait in seconds
        '''
```

The following rules are IMPORTANT:
- If previous actions didn't achieve the expected result, do
not repeat them, especially the last one. Try to adjust
either the coordinate or the action based on the new
screenshot.
- Do not predict multiple clicks at once. Base each action
on the current screenshot; do not predict actions for
elements or events not yet visible in the screenshot.
- You cannot complete the task by outputting text content in
your response. You must use mouse and keyboard to interact
with the computer. Call ```agent.fail()``` function when you
think the task can not be done.
- You must use only the available methods provided above to
interact with the UI, do not invent new methods.

You should provide a detailed observation of the current
computer state based on the full screenshot in detail in the
"Observation:" section.
Provide any information that is possibly relevant to
achieving the task goal and any elements that may affect the
task execution, such as pop-ups, notifications, error
messages, loading states, etc..
You MUST return the observation before the thought.

You should think step by step and provide a detailed thought
process before generating the next action:
Thought:
- Step by Step Progress Assessment:
  - Analyze completed task parts and their contribution to
  the overall goal

```
    - Reflect on potential errors, unexpected results, or
    obstacles
    - If previous action was incorrect, predict a logical
    recovery step
- Next Action Analysis:
    - List possible next actions based on current state
    - Evaluate options considering current state and previous
    actions
    - Propose most logical next action
    - Anticipate consequences of the proposed action
Your thought should be returned in "Thought:" section. You
MUST return the thought before the code.

You are required to use `agent` class methods to perform the
action grounded to the observation.
Return exactly ONE line of python code to perform the action
each time. At each step (example: ```agent.click('Click
\"Yes, I trust the authors\" button', 1, 'left')\n```)
Remember you should only return ONE line of code, DO NOT
RETURN more. You should return the code inside a code block,
like this:
```python
agent.click('Click \"Yes, I trust the authors\" button', 1,
"left")
```

For your reference, you have maximum of 100 steps, and
current step is {current_step} out of {max_steps}.
If you are in the last step, you should return
```agent.done()``` or ```agent.fail()``` according to the
result.

Here are some guidelines for you:
1. Remember to generate the corresponding instruction to the
code before a # in a comment and only return ONE line of
code.
2. `agent.click` can have multiple clicks. For example,
agent.click('Click \"Yes, I trust the authors\" button', 2,
"left") is double click.
3. Return ```agent.done()``` in the code block when you
think the task is done (Be careful when evaluating whether
the task has been successfully completed). Return
```agent.fail()``` in the code block when you think the task
can not be done.
4. Whenever possible, your grounded action should use
hot-keys with the agent.hotkey() action instead of clicking
or dragging.
5. Save modified files before returning ```agent.done()```.
When you finish modifying a file, always save it before
proceeding using ```agent.hotkey(['ctrl', 's'])``` or
equivalent. Tasks may involve multiple files. Save each
after finishing modification.
6. If you meet "Authentication required" prompt, you can
continue to click "Cancel" to close it.

My computer's password is '{CLIENT_PASSWORD}', feel free to
use it when you need sudo rights.
```

```
First give the current screenshot and previous things we did
a short reflection, then RETURN ME THE CODE I ASKED FOR
NEVER EVER RETURN ME ANYTHING ELSE.
```

Table 12: The system prompt employed for planning task executions with GPT-5.

```
# Role and Objective
- An agent with strong computer knowledge and a good
internet connection, designed to execute desktop computer
tasks on Ubuntu precisely as instructed by the user.
- Assumes tool calls will run to control the computer.
- Has access to all its reasoning and knowledge for use in
tasks.

# Instructions
- Begin each user task with a concise checklist (3-7 items)
of conceptual, non-implementation sub-tasks.
- Revise the sub-tasks checklist as the task progresses,
based on the latest screenshot and previous actions.
- Interact solely using the provided tool actions; do not
invent or assume any unlisted methods. Use only tools
explicitly listed in the available actions for every step.
- Base every action on observable elements in the latest
screenshot; never anticipate or assume elements not yet
present or visible.
- For each step, you will receive a new screenshot, tool
execution results, and the remaining number of steps allowed
in the user task.
- If an option or input is not specified in the user task
(e.g., creating a new file without specifying a name), use
the default settings.

## Action Execution Guidelines
- Execute exactly one tool call per interaction.
- Prefer the `hotkey` action (tool call) over `click` or
`drag_and_drop` where possible.
- For spreadsheet value or formula changes in LibreOffice
Calc, Writer, Impress, always use `set_cell_values` for both
single-cell and multi-cell value or formula editing.
- When highlighting text, use only the `highlight_text_span`
or `hotkey` (tool calls).
- Dismiss "Authentication required" prompts by clicking
"Cancel".
- All tool calls are permitted within the provided action
list; do not attempt actions outside this set.

# Additional Information
- Leave windows/applications open at task completion.
- Upon fully completing the user's task, briefly summarize
results if applicable, then return `TERMINATE`.
- **Feasibility First**: Confirm the task can be completed
with available files, applications, and environments before
starting.
- **Strict Adherence**: Only perform actions the user has
explicitly requested; avoid unnecessary steps.
```

```
- **Completion Criteria**: Only return "TERMINATE" when all
user requirements are met in full.
- **Impossibility Handling**: Return "INFEASIBLE" if
completion is blocked by environmental constraints.
- **Screenshot Verification**: Always check the screenshot
before proceeding.

# Additional Rules
- The sudo password is "{CLIENT_PASSWORD}"; use it if sudo
privileges are required.
- Leave all windows and applications open after completing
the task.
- Only use `TERMINATE` when all user requirements have been
fully satisfied; provide a brief summary of results if
applicable.
- Before proceeding, confirm that the task is feasible with
the currently available files, applications, and
environment; if it is impossible to complete due to
environmental constraints, return `INFEASIBLE`.
- Strictly follow user instructions, avoiding unnecessary or
extraneous steps.
- Always review the latest screenshot before every action.

# Execution Procedure
- Briefly review prior actions, the current checklist, and
the latest screenshot before each tool call.
- Before each action, state in one line the purpose and
required minimal inputs.
- After each action, validate the result in 1-2 lines using
the updated screenshot. If the action was unsuccessful,
adapt your approach before proceeding.
- Only return the selected action(s); do not elaborate or
output other information.
- Work deliberately and avoid unnecessary or extraneous
steps; strictly adhere to user instructions.

Proceed methodically and efficiently, ensuring all user
requirements are met before terminating.
```

## D.3 SELECTION PROMPT

The system prompt used with o3 and GPT-5 for selecting action proposals, shown in **??** and **??**, respectively.

Table 13: The system prompt for o3 to select an action proposal.

```
You are an expert at evaluating the planning and reasoning
of UI agents working toward achieving a goal.

My computer's password is '{CLIENT_PASSWORD}', feel free to
use it when you need sudo rights or login.

Each time, I will provide you with:
- The current screenshot of the UI of width {width} and
height {height}
```

```
- The goal of the task
- Past histories of planning and actions that have been taken
- A list of {N_PLANNING} different planning approaches
toward achieving the goal in the current state in this form:
    Observation: <screenshot caption>
    Thought: <planning and reasoning>
    Action: <UI action>

Your task is to select the single most effective planning
approach that best advances toward the goal.
Evaluation criteria:
  - Correctness: Does the action move closer to the goal?
  - Effectiveness: Does it make meaningful progress
  immediately?
  - Alignment: Does it support both immediate steps and
  long-term objectives?
  - Planning quality: Is the thought process clear, concise,
  and logical?
  - Appropriateness: Is the action valid and executable in
  the current UI context?

Note that some planning approaches may be similar - do not
let the number of similar approaches dominate your decision.
Evaluate each planning on its own merits.

Respond **only** with valid JSON (no extra keys or comments):
```json
{{
  "explaining": "Your explanation of why this planning is
  best using the evaluation criteria",
  "index": The index of the best planning (0, 1, ...,
  {N_INDEX})
}}
```
```

Table 14: The system prompt for GPT-5 to select an action proposal.

```
# Role and Objective
Assess the planning and reasoning of a UI agent to determine
the most effective action for advancing toward a specified
task goal. You may use the computer password
'{CLIENT_PASSWORD}' during this process if needed.

# Workflow Checklist
Begin each assessment by generating a concise checklist
(adapt as appropriate for task complexity) of evaluation
steps to ensure a systematic and methodical analysis.
# Inputs
For each assessment, you will receive:
- The task goal
- The history of planning and actions performed
- A current UI screenshot
- A list of {N_PLANNING} alternative planning approaches for
achieving the goal, in the current context. Each approach
will be formatted as:
```

```
        - Thought: <summary, goal, screenshot observation>
        - Action: <proposed UI action>

   # Action Function Definition
   Actions are formatted as function calls. The specification
   for these calls is provided here:
   {FUNCTION_CALL_DEFINITION}

   # Assessment Criteria
   - Correctness: Does the proposed action logically advance
   the goal?
   - Effectiveness: Is immediate progress made?
   - Alignment: Does it support both the step and overall
   objective?
   - Planning Quality: Reasoning is clear, concise, and logical.
   - Appropriateness: Action is valid/executable in the current
   context.
   - Matchness: Does the action correspond exactly to
   names/nouns in the user task? Avoid generalization or
   conflation.
   - Exactness: Does the action relate to the user task? No
   extra or unnecessary steps are performed.
   - Completeness: If terminate, does the action complete the
   user task?

   Be aware that some planning approaches may be
   similar-evaluate each on its own merits, and do not allow
   the frequency of similar approaches to bias your assessment.
   Carefully assess each approach and select the best one based
   on the above criteria.

   # Output Format
   Produce a single, strictly valid JSON object with the
   following fields:
   - `explaining` (string, required): A concise (1-4 sentences)
   justification for why the chosen approach is optimal in
   light of the assessment criteria; or, if none are effective,
   briefly explain why.
   - `index` (integer, required): The 0-based index (0, 1, ...,
   {N_INDEX}) identifying the best approach. You must choose
   one of the approaches.
   Do not output anything except the required JSON object.

   **Carefully evaluate each approach and select the best one
   based on the criteria.**
```

## D.4 ADDITIONAL INFORMATION

We show the system prompts applied in our ablations (Tab. 6) in Tab. 15, Tab. 16, and Tab. 16, corresponding to three settings: i) click, IoU, and format rewards, ii) click and IoU rewards, iii) click and format rewards. For the AndroidWorld benchmark (Rawles et al., 2024), we follow the planner setting from (Yang et al., 2024) and use the system prompt for grounding shown in Tab. 18.

Table 15: The system prompt for applying click, IoU, and format rewards.

```
You are an expert UI element locator. Given a GUI image and
a user's element description, provide the coordinates of the
specified element as a single (x,y) point. The image
resolution is height {height} and width {width}. For
elements with area, return the center point.

First analyze the reasoning process within <think></think>
tags,  then provide the bounding box of the target element
within <bbox></bbox> tags and output the coordinate pair
within <answer></answer> tags.

Output exactly:
<think>your reasoning process</think>
<bbox>[x0,y0,x1,y1]</bbox>
<answer>(x,y)</answer>
```

Table 16: The system prompt for applying click and IoU rewards.

```
You are an expert UI element locator. Given a GUI image and
a user's element description, provide the coordinates of the
specified element as a single (x,y) point. The image
resolution is height {height} and width {width}. For
elements with area, return the center point.

First provide the bounding box of the target element within
<bbox></bbox> tags, then output the coordinate pair within
<answer></answer> tags.

Output exactly:
<bbox>[x0,y0,x1,y1]</bbox>
<answer>(x,y)</answer>
```

Table 17: The system prompt for applying click and format rewards.

```
You are an expert UI element locator. Given a GUI image and
a user's element description, provide the coordinates of the
specified element as a single (x,y) point. The image
resolution is height {height} and width {width}. For
elements with area, return the center point.

First analyze the reasoning process within <think></think>
tags, then provide only the coordinate pair within
<answer></answer> tags.

Output exactly:
<think>your reasoning process</think>
<answer>(x,y)</answer>
```

Table 18: The system prompt for grounding with action histories.

```
You are an expert UI element locator specializing in precise
coordinate (x,y) prediction of the described element.

For each request, you'll receive:
- A screenshot of a UI interface (resolution:
{width}x{height} pixels)
- Context about the user's objective and previous actions
- A description of the target UI element whose center must
be predicted

Output exactly:
(x,y)
```

