# OpenReview forum: "GTA1: GUI Test-time Scaling Agent"
_ICLR.cc/2026/Conference — ICLR 2026 Poster_

### Official Review · Reviewer_UGQ4 · 2025-10-17

**Soundness:** 3
**Presentation:** 3
**Contribution:** 3
**Rating:** 6
**Confidence:** 5

**Summary:**

This paper introduces the GUI Testing-time Scaling Agent (GTA1), which aims to solve challenges in planning and action localization for GUI agents. Through a testing-time scaling strategy, it samples multiple candidate action proposals at each step, from which a judgment model selects the optimal solution. This is combined with a reinforcement learning-based localization model that directly predicts interaction coordinates and is optimized via click rewards, leading to effective performance improvement.
Experimental results show that GTA1 achieves state-of-the-art performance on multiple benchmarks. For instance, on the ScreenSpot-Pro benchmark, the 7B model reached an accuracy of 50.1%, surpassing UGround-72B's 34.5%. On the OSWorld benchmark, GTA1-7B achieved a task success rate of 45.2%, outperforming all compared methods. These results demonstrate the effectiveness and robustness of the proposed method.

**Strengths:**

1.GTA1 effectively addresses the planning and localization challenges for GUI agents through a testing-time scaling strategy and a reinforcement learning-based localization model, achieving state-of-the-art performance across multiple benchmarks.

2.Extensive experiments have validated the effectiveness and generalizability of the method.

**Weaknesses:**

It is noted in the manuscript that the foundational model for GTA1-7B is UI-TARS-1.5-7B, while for GTA1-32B it is OpenCUA-32B. This raises the question of how the performance would be affected if qwen2.5-vl or qwen3-vl were used as the base models instead. Further experimentation is required to exhibit the superior generalization capabilities of the proposed methodology.

**Questions:**

1.I have observed that the training data originates from Aria-UI-Web, OmniACT, UI Vision, Widget Caption, and OSAltas-Desktop. Could you elaborate on the rationale for selecting these particular datasets? Furthermore, what considerations were made in terms of the volume and heterogeneity of the data?

2.it is stated that the training dataset underwent a filtering process based on the quality of bounding boxes. Could you provide the specific quantities of the dataset both prior to and subsequent to this filtering procedure?

3.This manuscript evaluates performance on benchmarks such as OS-World and WindowsAgentArena. What was the reasoning for not incorporating agent data into the training set? Additionally, could you compare the potential efficacy of a unified model that performs both planning and grounding, versus the current methodology which involves test-time scaling for planning followed by a separate grounding step?

4.It is mentioned that 70k data samples were utilized for training. Is it anticipated that performance would continue to enhance with a further increase in the scale of the data? I wonder if there is a scaling law here?

---

> ### Author Response · Authors · 2025-11-26
> **Response [1]**
>
> We thank the reviewer for recognizing the effectiveness of GTA1 and acknowledging the breadth of our experimental evaluation. Below, we provide detailed responses to the remaining concerns.
>
> > **W1**: This raises the question of how the performance would be affected if qwen2.5-vl or qwen3-vl were used as the base models instead.
>
> **A**: We further experiment with qwen2.5-vl-32B and qwen2.5-vl-72B  for grounding performance.  The results across multiple grounding benchmarks are summarized in the table below.
>
> |   Dataset                | ScreenSpot-V2 | ScreenSpot-Pro | OSWorld-G | OSWorld-G-Refined  |
> |--------------------------|---------------|----------------|-----------|--------------------|
> | Qwen2.5-VL-32B (baseline)| 91.9          |    47.6        | 46.5      | 59.6               |
> | Qwen2.5-VL-32B           | 93.2          |    53.6        | 54.4      | 61.9               |
> | Qwen2.5-VL-72B (baseline)| 94.0          |    53.3        | 54.1      | 62.2               |
> | Qwen2.5-VL-72B           | 94.8          |    58.4        | 57.6      | 66.7               |
>
>
> Overall, our method consistently achieves state-of-the-art–level grounding performance across different backbone architectures, demonstrating the robustness and generality of our approach.
>
> > **Q1**:  Could you elaborate on the rationale for selecting these particular datasets? Furthermore, what considerations were made in terms of the volume and heterogeneity of the data?
>
> **A**: Our training dataset is composed of diverse GUI datasets covering OS, web, and mobile domains to ensure broad coverage of interface types and tasks. After filtering for quality, we sample from these datasets to construct a 70K-example training set, as summarized below.
>
>
> | Domain   | Dataset             | % of 70K |
> |:--------:|---------------------|----------|
> |          | OS-Atlas-Desktop    | 57.3%    |
> |  OS      | OmniACT             | 2.6%     |
> |          | UI Vision           | 5.7%     |
> | Web      | Aria-UI-Web         | 14.2%    |
> | Mobile   | Widget Caption      | 20.2%    |
>
>
>
>
> > **Q2**: It is stated that the training dataset underwent a filtering process based on the quality of bounding boxes. Could you provide the specific quantities of the dataset both prior to and subsequent to this filtering procedure?
>
>
> **A**:  Among all the datasets used, only Aria-UI-Web and OS-Atlas-Desktop required filtering due to noise introduced by the HTML and A11y parsers. After applying our bounding-box quality filter, 33.1% of the Aria-UI-Web samples and 34.4% of the OS-Atlas-Desktop samples are removed.

---

> ### Author Response · Authors · 2025-11-26
> **Response [2]**
>
> > **Q3**:  What was the reasoning for not incorporating agent data into the training set? Additionally, could you compare the potential efficacy of a unified model that performs both planning and grounding, versus the current methodology which involves test-time scaling for planning followed by a separate grounding step?
>
> **A**: Our training objective is centered specifically on GUI grounding, whereas the agent datasets focus on high-level action planning. Incorporating agent data would therefore introduce a potential distribution mismatch with the low-level grounding. Therefore, we chose not to include agent trajectories in the training set.
>
> To evaluate the potential of a unified model,, we conduct two set experiments for study on the OSWorld benchmark. Firstly, we experiment using o3 as the planner, and the success rate results (%) are summarized below.
>
> | With Test-time Scaling | With GTA1-7B Grounding | Success Rate |
> | ---------------------- | ---------------------- | ---------------- |
> | No                     | No                     |    20.1          |
> | Yes                    | No                    |    22.4          |
> | No                     | Yes                    |    43.4          |
> | Yes                    | Yes                    |    45.2          |
>
>
> Secondly, we experiment using the UI-TARS-7B as the planner, with the results below.
>
>
> | With Test-time Scaling | With GTA1-7B Grounding | Success Rate  |
> | ---------------------- | ---------------------- | ---------------- |
> | No                     | No                     |    22.7          |
> | No                     | Yes                    |    21.2          |
> | Yes                    | No                    |    27.9          |
> | Yes                    | Yes                    |    30.1          |
>
>
> Consistently, the combination of our test-time scaling strategy with the grounding model yields better performance compared to the reverse setting.
> > **Q4**: Is it anticipated that performance would continue to enhance with a further increase in the scale of the data?
>
>
> In our preliminary experiments, we observe consistent improvements as the amount of training data increases. Expanding the dataset with more diverse and higher-quality GUI grounding samples is a promising direction, as it further shapes the model’s ability to handle diverse interface patterns. We believe that additional data would continue to scale the model’s performance.

---

### Official Review · Reviewer_88ea · 2025-10-24

**Soundness:** 3
**Presentation:** 3
**Contribution:** 3
**Rating:** 8
**Confidence:** 3

**Summary:**

GTA1 is a GUI agent that improves both plan selection and precise clicking on complex screens. At each step, it samples multiple action proposals and uses a judge model to pick the best one, trading extra compute for better decisions. For grounding, it trains a simple RL model that directly predicts click coordinates with a click reward, achieving SOTA results on ScreenSpot-Pro, ScreenSpot-V2, and OSWorld-G, and strong task success on OSWorld and WindowsAgentArena. The key contributions are test-time scaling for robust planning, a minimal RL recipe for grounding, and an analysis showing “thinking” isn’t needed in static UIs but can help in dynamic ones.

**Strengths:**

1. The paper tackles two concrete pain points—plan selection and precise grounding, and pairs them with two simple fixes: test-time scaling for planning and a minimal RL recipe for click grounding. The framing is clear and the approach maps tightly to the problems.

2. Test-time scaling actually pays off. Sampling K proposals per step and using a judge consistently boosts success while also cutting wall-clock via concurrent sampling.

3. Directly rewarding clicks inside the target element keeps the training signal simple and on-task, yet reaches SOTA on ScreenSpot-Pro, ScreenSpot-V2, and OSWorld-G. The simplicity is a strength.

4. The paper shows that explicit reasoning isn’t needed for static UIs and can even hurt, while helping in dynamic settings when trajectories/objectives are involved. This nuance helps practitioners know when to spend tokens on reasoning.

5. Results cover both grounding and full agent task execution (OSWorld and WindowsAgentArena), and scaling tests generalize across agents, horizons, and K settings, with qualitative analyses of cascading failures. The breadth makes the claims more convincing.

**Weaknesses:**

1. Test-time scaling samples K proposals each step and uses a judge, which can raise token and inference costs even if concurrent sampling cuts wall-clock time. The paper shows speedups and success gains but does not report detailed cost curves (tokens, latency) across K and horizons.

2.  The paper observes little average gain from “thinking” and attributes sample-wise differences to training instability. “Thinking helps only in dynamic UIs” needs stronger controls. The authors can test different “thinking” budgets, report variance to separate instability from genuine reasoning effects, and clarify when trajectories/objectives trigger reliable wins.

**Questions:**

1. Experiments use o3/GPT-5 as planners and a multimodal LLM judge. Can you test how performance transfers to an open-source stack?

2. Could you report the judge’s pairwise selection accuracy and show how judge errors propagate to end-task failure?

3. Can you share success–vs–cost curves (tokens, latency, $) across K and horizon, and results for adaptive policies (early exit, dynamic K per step)?

---

> ### Author Response · Authors · 2025-11-26
> **Response**
>
> We thank the reviewer for acknowledging that the simplicity of our components is one of the strengths and for recognizing the breadth of our experiments. Below, we provide point-by-point responses to the raised concerns and suggestions.
>
> > **W1** and **Q3**: Cost curves (tokens, latency) across K and horizons.
>
> **A**: Thank you for the suggestion. We have added the cost curves showing the number of tokens (which directly correlates with latency) across different values of $K$ and planning horizons in Appendix A.2. We observe a consistent trend: as token usage increases, the model achieves better performance.
>
> > **W2**: The paper observes little average gain from “thinking” and attributes sample-wise differences to training instability. “Thinking helps only in dynamic UIs” needs stronger controls. The authors can test different “thinking” budgets, report variance to separate instability from genuine reasoning effects, and clarify when trajectories/objectives trigger reliable wins.
>
> **A**:  To further examine the claim that “thinking” helps primarily in dynamic UIs, we provide additional evaluations and analyses on the AndroidWorld benchmark. We have trained three  “thinking” models , and their obtained success rates are 44.0%, 42.2%, and 47.0%, with a mean of 44.4% and a variance of 1.98%. In comparison, three “non-thinking” models achieve success rates of 38.8%, 34.9%, and 39.2%, with a mean of 37.6% and a standard deviation of 1.94%. It shows a consistent gain from “thinking”.
> > **Q1**: Open-source judge
>
> **A**:  Thank you for the suggestion. We experiment with the  $\text{Qwen3-VL-8B-Thinking}$  as the judge and consistently observe performance improvements as the number of action proposals $K$ increase.
> For a 15-step horizon, with an o3 judge we obtain 25.7,  26.1, and 27.9 task success rates (%) for $K \in \{8,16,32\}$, respectively; the baseline $K=1$ achieves 22.7. Following the setup, using $\text{Qwen3-VL-8B-Thinking}$ as the judge yields 27.0, 27.4, and 28.7 for $K \in \{8,16,32\}$. Consistently, increasing $K$ of our test-time scaling improves task success.
>
> > **Q2**:  Could you report the judge’s pairwise selection accuracy and show how judge errors propagate to end-task failure?
>
> **A**:  We randomly sampled 100 steps from task executions and manually evaluated whether the judge selected the correct action. In 19 of these steps, the planner fail to propose the correct action at all, which necessarily makes the judge’s choice incorrect. Excluding those cases, the o3 judge achieves a 91.4% selection accuracy. Furthermore, in Appendix A.3, we provide a failure case study. In many cases, none of the sampled action proposals are correct, causing the judge to select an invalid action. These errors, arising from the planner providing no valid candidates, propagate through the task execution and ultimately lead to task failure.

---

### Official Review · Reviewer_9sM6 · 2025-10-31

**Soundness:** 2
**Presentation:** 3
**Contribution:** 2
**Rating:** 4
**Confidence:** 4

**Summary:**

The paper introduces a GUI agent that works in two stages: planning and grounding. The proposed solution improves each stage differently:
1. the planning stage through test-time scaling: sampling K times at each step and asking a judge to select the "best" one.
2. the grounding stage by fine-tuning a model using GRPO

**Strengths:**

- strong empirical results on relevant benchmarks
- simple method
- ablations and discussion on the efficacy of the different reward signals (for thinking, for location)

**Weaknesses:**

- the dataset curation step should be part of the "algorithm", otherwise the comparison with other methods is not
- reduced novelty
- no ablation to understand the improvement brought by each of the two components in the agent task execution scenarios

**Questions:**

How is the training dataset built from the collection mentioned in the paper?

---

> ### Author Response · Authors · 2025-11-26
> **Response**
>
> We thank the reviewer for recognizing that our simple method achieves strong empirical results on relevant benchmarks. Below, we provide a point-by-point response addressing the reviewer’s concerns.
>
> > **W1** and **Q1**: The dataset curation step should be part of the "algorithm", otherwise the comparison with other methods is not.
>
> **A**: Thank you for the suggestion. We would like to clarify that we are not entirely certain about the concern. If it pertains to requesting more details about the datasets, we have revised the paper to expand the description of our data curation steps. Kindly refer to Appendix B for the detailed dataset curation process.
>
> Specifically, our seed dataset contains Aria-UI-Web, OmniACT, UI Vision, Widget Caption, and OS-Atlas-Desktop. We then apply the data cleaning algorithm described in Sec. 3.1 with $\tau = 0.3$, and the retained dataset is sampled as summarized in the table below to form our 70K training dataset.
>
> | Domain   | Dataset             | % of 70K |
> |:--------:|---------------------|----------|
> |          | OS-Atlas-Desktop    | 57.3%    |
> |  OS      | OmniACT             | 2.6%     |
> |          | UI Vision           | 5.7%     |
> | Web      | Aria-UI-Web         | 14.2%    |
> | Mobile   | Widget Caption      | 20.2%    |
>
>
> Please let us know if further clarification is needed.
>
> > **W2**:  reduced novelty
>
> **A**: We would like to emphasize that our contributions are encapsulated into two folds:
>
> - Test-time scaling of planning. Our planner samples multiple candidate proposals at each step, and a multimodal LLM selects the most contextually appropriate option, enabling effective short-horizon exploration without full rollouts. This strategy consistently improves performance. For example, with test-time scaling, the task success rate of UI-TARS-1.5-7B increases from 24.6 to 35.8 under a 50-step horizon (Fig. 4).
>
> - Post-training of grounding model. We show that GRPO objective is aligned directly with the grounding task, rather than relying on textual “thinking”, leading to the state-of-the-art grounding models across benchmarks. For example, GTA1-32B achieves 63.6% accuracy on the ScreenSpot-Pro dataset, outperforming the best proprietary model UI-TARS-1.5 by 2%.
>
> Taken together, these two components represent meaningful technical contributions that extend beyond prior work.
>
> If the reviewer has specific concerns about our model components or about novelty relative to particular prior methods, we would be happy to clarify further.
>
> > **W3**: no ablation to understand the improvement brought by each of the two components in the agent task execution scenarios.
>
> **A**: We have provided additional ablation studies to further show the performance of our component, as described below:
>
> - Without applying any time-scaling, using o3 alone (no GTA1 grounding) yields a 20.1 task success rate, while adding our grounding model improves the performance to  43.4 .
>
> - With test-time scaling, the performance of o3 alone (without GTA1 grounding) increases to 22.4%.
> With our grounding model applied, the performance further scales to 45.2%.
>
> Consistently, both our test-time scaling and GTA grounding contribute to improved task performance, demonstrating that each component provides complementary benefits in agent task execution.

---

### Official Review · Reviewer_p5q9 · 2025-11-01

**Soundness:** 3
**Presentation:** 2
**Contribution:** 2
**Rating:** 4
**Confidence:** 4

**Summary:**

The paper’s main contributions are (i) a test-time scaling strategy that selects the next action and (ii) a GRPO-trained GTA model for coordinate grounding. Specifically, at each step, the planner samples multiple candidate actions, and a multimodal LLM judge selects the best one. For actions requiring coordinates, a GTA model trained with GRPO provides more precise grounding. The GTA grounding model achieves SOTA on grounding benchmarks (ScreenSpot-Pro, ScreenSpot-V2, OSWorld-G). On real OSWorld tasks, o3 with test-time scaling + GTA1-7B grounding attains a 45.2% success rate with a 100-step cap, surpassing CUA o3 at 42.9% with a 200-step cap.

**Strengths:**

- Strong empirical results across both grounding datasets and interactive environments, reaching SOTA on multiple benchmarks.
- The two-stage (planning+grounding) framework makes it straightforward to switch Planner/Judge modules or upgrade the grounding model.
- The light-weight data cleaning pipeline can be integrated with existing data pipelines.

**Weaknesses:**

- The description of test-time scaling is under-specified. For example, when the judge "picks the best candidate", is the output a score over candidates, or a rewritten action? Some results (e.g., Table 1) could be simplified to improve readability by moving some baselines to the appendix.
- Evaluations of test-time scaling (TTS) focus on ablations against the same agent without TTS. At least there should be equal-compute comparisons (e.g., self-consistency / majority voting / alternative sampling strategies).
- There are no confidence intervals. Reported gains over an OpenAI o3 baseline are ~2 points (Table 4). Without multiple runs and confidence intervals, statistical significance is unclear.
- The results of larger models are missing. GTA1-7B with GPT-5 and 32B configurations are "unavailable". If this is due to API cost, a proxy study (e.g., replaying 7B trajectories with a 32B grounding model) would strengthen the insights of the experiments. Likewise, results for open-source planners with test-time scaling + GTA grounding would be valuable.
- The paper appears to lack OSWorld ablations isolating the impact of the GTA grounding model.
- The proposed method largely adapts known ideas. Test-time scaling here is similar to best-of-N, and the GTA model is based on GRPO + data cleaning + reward shaping. It is reasonable that these techniques can boost performance, so more analysis and insights are required in the experiments (see suggestions above).

**Questions:**

- Under matched compute, how much does your test-time scaling improve over other test-time strategies?
- Does the judge score candidates or generate a new action, and how sensitive are results to this choice?
- If you increase the number of action proposals beyond 32, do gains continue?
- What is the performance delta on OSWorld with and without the grounding model? Please provide ablations quantifying its impact.
- How do open-source models perform under your test-time scaling, with and without the GTA grounding model? It would help to report accuracy-cost curves and confidence intervals for these settings.

---

> ### Author Response · Authors · 2025-11-26
> **Response [1]**
>
> We thank the reviewer for recognizing our empirical results across different grounding datasets and interactive environments, as well as our lightweight data-cleaning pipeline. We also appreciate the constructive suggestions. Please find our responses below.
>
> > **W1**: The description of test-time scaling is under-specified. Some results (e.g., Table 1) could be simplified.
>
> **A**:  We address this concern in two parts.
>
> - Description of test-time scaling.  Our test-time scaling procedure presents the candidate proposals as a multiple-choice set to the judge, who selects the final action by outputting the index of the chosen option.   (Sec. 3.1, Line 185; Fig. 2). Additional implementation details, including the system prompt used by the judge, are provided in the Appendix D.3.
> - Clarification on Table 1.  We have revised Tab. 1 to improve clarity by removing several poorly performing methods, resulting in a cleaner and more focused comparison.
>
> > **W2** and **Q1**:  Equal-compute comparisons for test-time scaling.
>
> **A**:  We thank the reviewer for the insightful comments. To provide an equal-compute comparison, we evaluate a majority-voting baseline under the same setup as Fig. 4 under a 15-step horizon. The results are summarized below.
>
> | Method         | $K=1$ | $K=8$ | $K=16$ | $K=32$ |
> |----------------|-----|-----|------|------|
> | Ours           | 22.7| 25.7| 26.1 | 27.9 |
> | Majority Vote  | 22.7| 23.6| 24.1 | 23.8 |
>
>
> Across all scaling factors, our method consistently outperforms majority voting, which does not reliably improve with larger $K$. In majority voting, the most frequent action is selected for execution. This highlights the advantage of our judge-based test-time scaling strategy.
>
> > **W3**: Reported gains over an OpenAI o3 baseline are ~2 points (Table 4). Statistical significance is unclear.
>
> **A**: Kindly note the ~2 points gain referenced in the review is computed against a CUA-specific o3 baseline (i.e., CUA o3) that is not publicly accessible (no API key available). To ensure a fair and transparent comparison, we have added an o3 baseline, which uses o3 as an end-to-end model for both planning and grounding, to Tab. 4. Against the o3 baseline (20.1 success rate), our method achieves an improvement of 25.1% points in task success rate.
>
> For the statistical significance, we thank the reviewer for the valuable suggestion. Due to time limitation, we conduct experiments using the officially small test set split of OSWorld-Verified. Three test-time scaling runs are performed, with task success rates summarized below.
>
>  Model | Run 1 | Run 2 | Run 3 | Avg ± STD    |
> ----------------- | ----- | ----- | ----- | -------------|
> o3                | 53.9  | 51.3  | 53.9  |  53.0 ± 1.23 |
>
> The standard deviation (STD) is computed over the three runs to reflect the variability of test-time scaling performance. These results demonstrate the consistently strong performance of our method. When expanding to the full test set, we would anticipate an even more stable standard deviation.
>
>
> > **W4** GTA1-7B with GPT-5 and 32B configurations are "unavailable".
>
>
> **A**: OSWorld is no longer maintained and has been updated to OSWorld-Verified for [more reliable evaluations](https://xlang.ai/blog/osworld-verified). For fair comparison with prior works that are benchmarked on OSWorld, we report results on the original benchmark while also providing updated results on OSWorld-Verified.
>
>
> To further strengthen the comparison, we have additionally included an o3 baseline. Our “o3 + GTA1-7B” configuration outperforms this baseline by 25.1% on OSWorld and 30.1% on OSWorld-Verified.
> > **W5** and **Q4**:  The paper appears to lack OSWorld ablations isolating the impact of the GTA grounding model.
>
> **A**: We have expanded our ablation study on the OSWorld by exploring using o3 alone for task execution under a 100-step horizon.
>
> | With Test-time Scaling | With GTA1-7B Grounding | Success Rate  |
> | ---------------------- | ---------------------- | ---------------- |
> | No                     | No                     |    20.1          |
> | Yes                    | No                    |    22.4          |
> | No                     | Yes                    |    43.4          |
> | Yes                    | Yes                    |    45.2          |
>
>
> These results demonstrate the substantial performance gains contributed by both our grounding model and the test-time scaling strategy.

---

> ### Author Response · Authors · 2025-11-26
> **Response [2]**
>
> > **W6**:  Test-time scaling here is similar to best-of-N.
>
> **A**:  Our test-time scaling strategy is significantly different from the best-of-N approach, as summarized below:
> -  Best-of-N executes the full task multiple times in the environment. However, some environment states are infeasible to reset (e.g., after deleting or removing files), making repeated execution unreliable or impossible (as discussed in Lines 44–46).
> - Our method samples multiple candidate plans and selects the most appropriate one at each step of task execution. This stepwise decision-making process enables the agent to navigate complex user tasks efficiently by scaling the test-time compute, without altering the environment state in the planning phase.
> Therefore, we believe that our approach provides an effective and environment-safe test-time scaling mechanism that preserves planning flexibility while maintaining execution correctness, which are the capabilities that are absent in the best-of-N approach.
>
> > **Q2**:  Does the judge score candidates or generate a new action, and how sensitive are results to this choice?
>
> **A**: Our test-time scaling method presents the candidate proposals as a multiple-choice set to the judge, who selects the best prediction by outputting the corresponding candidate index (Sec. 3.1, Line 185; Fig. 2). Additional implementation details, including the system prompt used by the judge, are provided in Appendix D.3.
>
> Regarding sensitivity, refer to the response to **W3** for more details.  Overall, our method exhibits a standard deviation of approximately 1.2%, computed over three runs to capture the variability of test-time scaling performance. These results indicate that our approach maintains consistently strong and stable performance across evaluations.
>
> > **Q3**: If you increase the number of action proposals beyond 32, do gains continue?
>
> **A**: We do not observe further gains when increasing the number of action proposals beyond 32.
>
> > **Q4**: How do open-source models perform under your test-time scaling, with and without the GTA grounding model? It would help to report accuracy-cost curves and confidence intervals for these settings.
>
> **A**:  Our method enables open-source models to scale their performance, irrespective of whether our GTA1 grounding is used. We observe that increasing the scaling factor ($K$) consistently improves task success rates. Following Fig. 4, which evaluates the open-source UI-TARS model under our test-time scaling strategy, we additionally experiment with using the end-to-end UI-TARS model as the planner and GTA1-7B as the grounder. The results are summarized below:
>
> | Method           | $K=1$ | $K=8$ | $K=16$ | $K=32$ |
> |------------------|-----|-----|------|------|
> | w/o our grounding| 22.7| 25.7| 26.1 | 27.9 |
> | w/ our grounding | 21.2| 28.4| 29.5 | 30.1 |
>
> Our grounding model consistently improves performance as $K$ increases, demonstrating that the benefits of test-time scaling extend to open-source planners as well.
>
> Regarding the requested accuracy–cost curves, we clarify that the value of $K$ is directly proportional to token cost. Following the suggestion, we have added a token-cost figure in Appendix A.2 to illustrate this relationship more explicitly.
>
> We kindly request the reviewer to consider raising the score if the clarifications above address the concern.

---

### Comment · Area_Chair_bnuV · 2025-11-27
**Reviewers: please read the rebuttals and make necessary edits**

Dear Reviewers,

Please read the authors' rebuttals and make necessary edits to your reviews.

Best,

AC

---

### Author Response · Authors · 2025-12-02
**Rebuttal Summary**

Dear AC,

The initial scores for our submission are **4**, **4**, **8**, **6**. There has been no further engagement from the reviewers before the freeze, despite our comprehensive response.

We have effectively addressed the reviewers' concerns and strengthened the paper through extensive new experiments. Below is a summary of the key feedback and our responses, presented in the order of the reviewers.

**Reviewer p5q9 (Score: 4)**

- **Feedback Summary**: The reviewer requested equal-compute comparisons (e.g., majority voting) for the test-time scaling, questioned the statistical significance of results, and sought clarification on performance gains relative to baselines.

- **Our Response Summary:**
    - **Equal Compute**: We introduced a **Majority Voting** baseline under matched compute constraints. Our method consistently outperformed Majority Voting, verifying the efficacy of our judge-based selection.
    - **Significance & Baselines**: We clarified that the cited baseline (CUA o3) was inaccessible and added a standard **o3 baseline**. Our method achieves a significant improvement over this baseline. We also provided statistical analysis showing a low standard deviation across runs to confirm stability.

**Reviewer 9sM6 (Score: 4)**

- **Feedback Summary**: The reviewer raised concerns about novelty, requested details on dataset curation, and asked for ablations to isolate the impact of the planning and grounding components .

- **Our Response Summary**:

    - **Clarification**: While the reviewer's comment regarding "reduced novelty" was not specific, we endeavored to address it by clarifying our distinct technical contributions: a step-wise test-time scaling strategy (avoiding full rollouts) and a GRPO-aligned grounding objective.
    - **Ablations**: We provided granular results showing that both components are critical and complementary, demonstrating that the combination yields superior performance compared to using either in isolation.
    - **Dataset**: We expanded Appendix B to detail the data curation and filtering process.

**Reviewer 88ea (Score: 8)**

- **Feedback Summary**: The reviewer strongly endorsed the work, noting that the test-time scaling "actually pays off". They requested cost curves (tokens/latency), further validation of the "thinking" hypothesis in dynamic UIs, and performance data using open-source judges.

- **Our Response Summary**:
    - **Cost & Open Source Model**: We added token-cost curves to Appendix A.2. We also validated our method using an open-source judge (Qwen3-VL-Thinking), showing consistent performance scaling as action proposals increased.
    - **"Thinking" Validation**: We provided new experiments on AndroidWorld showing that "thinking" models consistently outperform non-thinking variants, confirming the utility of reasoning in dynamic environments.

**Reviewer UGQ4 (Score: 6)**

- **Feedback Summary**: The reviewer praised the method's robustness and SOTA performance, and asked about generalization to other base models (e.g., Qwen) and the efficacy of a unified model versus our two-stage approach.

- **Our Response Summary**:
    - **Generalization**: We applied our grounding recipe to **Qwen2.5-VL** (32B & 72B), achieving better performance compared to the baselines .
    - **Unified vs. Two-Stage**: We empirically demonstrated that our two-stage approach (Planner + Grounder) consistently outperforms unified end-to-end models.

Best,

The Authors

---

### Meta-Review · Area_Chair_XX3x · 2026-01-08

**Summary:**

The paper proposed GTA1, a GUI agent that achieves SOTA performance via a combination of an efficient action selection and an improved action grounding mechanism that relies on GRPO.

The reviewers' biggest concerns were:
- Missing ablations, including those on OSWorld, isolating the effect of each of the system's components on performance
- The contribution being largely an aggregation of known techniques, limiting the novelty
- Missing experiments that would use Qwen-VL model series as base models for this agent

The authors have conducted a lot of additional experiments for the rebuttal and clarified which contributions they consider novel.

In the metareviewer's opinion, the additional experiments directly addressed many of the reviewers' concerns. Contribution-wise, the metareviewer found the *algorithmic* and *methodological* novelty indeed limited. However, it's just a different type of submission, an "agentic system" paper, whose contributions consist in extensive empirical analysis identifying what works, *combined with* adaptations of known algorithmic techniques. In the metareviewer's opinion, given this work's empirical thoroughness (especially after the rebuttal) and the resulting agent's strong performance, other people building CUA agents will find many useful insights in it.

**Reviewer Concerns:**

Please see above.

**Reviewer Scores:**

No reviewer responded to the rebuttals before the freeze.

Reviewers p5q9 and 9sM6 gave a score of 4, and, other than perhaps the contribution originality, the rebuttal directly addressed their other concerns, so they would potentially have raised their score to 5 or 6.

Reviewer 88ea gave a score of 8 and while the rebuttal quite directly addressed their concerns too, the score would have probably remained the same.

Reviewer UGQ4 gave a 6 and it's possible they would have increased it to 7 after the rebuttal or left it at 6.

---

### Decision · Program_Chairs · 2026-01-26

Accept (Poster)